# Spectral Perturbation Bounds for Low-Rank Approximation with Applications to Privacy

**Phuc Tran**
VinUniversity

**Nisheeth K. Vishnoi**[*]
Yale University

**Van H. Vu**
Yale University

## Abstract

A central challenge in machine learning is to understand how noise or measurement errors affect low-rank approximations—particularly in the spectral norm. This question is especially important in differentially private low-rank approximation, where one aims to preserve the top-$p$ structure of a data-derived matrix while ensuring privacy. Prior work often analyzes Frobenius norm error or changes in reconstruction quality, but these metrics can over- or under-estimate true subspace distortion. The spectral norm, by contrast, captures worst-case directional error and provides the strongest utility guarantees. We establish new high-probability spectral-norm perturbation bounds for symmetric matrices that refine the classical Eckart–Young–Mirsky theorem and explicitly capture interactions between a matrix $A \in \mathbb{R}^{n \times n}$ and an arbitrary symmetric perturbation $E$. Under mild eigengap and norm conditions, our bounds yield sharp estimates for $\|(A + E)_p - A_p\|$, where $A_p$ is the best rank-$p$ approximation of $A$, with improvements of up to a factor of $\sqrt{n}$. As an application, we derive improved utility guarantees for differentially private PCA, resolving an open problem in the literature. Our analysis relies on a novel contour bootstrapping method from complex analysis and extends it to a broad class of spectral functionals, including polynomials and matrix exponentials. Empirical results on real-world datasets confirm that our bounds closely track the actual spectral error under diverse perturbation regimes.

## 1 Introduction

Low-rank approximation is a foundational technique in machine learning, data science, and numerical linear algebra, with applications ranging from dimensionality reduction and clustering to recommendation systems and privacy-preserving data analysis [1, 4, 5, 14, 21, 23, 24, 42, 45]. A common setting involves a real symmetric matrix $A \in \mathbb{R}^{n \times n}$, such as a sample covariance matrix derived from high-dimensional data. Let $\lambda_1 \geq \cdots \geq \lambda_n$ denote the eigenvalues of $A$, with corresponding orthonormal eigenvectors $u_1, \ldots, u_n$. The best rank-$p$ approximation of $A$ is denoted by $A_p := \sum_{i=1}^{p} \lambda_i u_i u_i^\top$. This approximation solves the optimization problem $A_p = \arg\min_{\text{rank}(B) \leq p} \|A - B\|$, where the norm can be any *unitarily invariant norm* [7, 10]. In particular, $A_p$ minimizes both the *spectral norm* $\|\cdot\|$, measuring worst-case error, and the *Frobenius norm* $\|\cdot\|_F$, measuring average deviation.

In many applications, the matrix $A$ is not directly available—it may be corrupted by noise, compressed for efficiency, or randomized to preserve privacy. A standard model introduces a symmetric perturbation $E$, yielding the observed matrix $\tilde{A} := A + E$. The approximation $\tilde{A}_p$, computed from $\tilde{A}$, is often used in downstream learning and inference. This leads to a central question: *How does the perturbation $E$ affect the top-$p$ approximation $A_p$?* Understanding the deviation $\|\tilde{A}_p - A_p\|$ is critical for ensuring the reliability and robustness of low-rank methods under noise.

---

[*]Alphabetical order. Correspondence to `nisheeth.vishnoi@gmail.com`.

39th Conference on Neural Information Processing Systems (NeurIPS 2025).

**Motivating application: differential privacy.** The stability under perturbations is especially important when the matrix $A$ encodes *sensitive information*, such as user behavior or medical data. In such settings, even low-rank approximations of $A$ can inadvertently leak private information [6]. To address this risk, differential privacy (DP) [14] has become the standard framework for designing privacy-preserving algorithms. Several mechanisms have been developed to release private low-rank approximations while satisfying DP guarantees [8, 9, 15, 25, 29, 31, 34, 39]. A canonical method, introduced in [15], adds a symmetric noise matrix $E$ with i.i.d. Gaussian entries to the input matrix $A$, yielding the perturbed matrix $\tilde{A} = A + E$. The algorithm then releases $\tilde{A}_p$ as the privatized output. The *utility* of such mechanisms is typically assessed by comparing $\tilde{A}_p$ to the ideal (non-private) approximation $A_p$. Two standard metrics are: (1) the *Frobenius norm error* $\|\tilde{A}_p - A_p\|_F$, and (2) the *change in reconstruction error* $|\|A - A_p\|_\star - \|A - \tilde{A}_p\|_\star|$, which measures how much the quality of low-rank approximation degrades due to noise, for a norm $\|\cdot\|_\star$ [3, 11, 15, 29]. These metrics offer insight into the effect of noise on overall variance or total reconstruction error. However, as we explain next, they may fail to capture *worst-case directional misalignment*, which is often critical for downstream tasks and algorithmic guarantees.

**Limitations of existing utility metrics.** The Frobenius norm error and reconstruction error may not be appropriate in applications that rely on the geometry of the top-$p$ eigenspace. In particular, the Frobenius norm may *overestimate* the impact of noise by up to a factor of $\sqrt{p}$ when the perturbation $E$ lies largely in directions orthogonal to the top-$p$ subspace. The reconstruction error metric can *underestimate* subspace deviation—sometimes dramatically. In some cases, it remains small (or even zero) despite substantial rotation in the top-$p$ eigenspace. (See Sections B for concrete illustrations.) These limitations motivate the use of the *spectral norm* $\|\tilde{A}_p - A_p\|$, which captures the *worst-case* directional deviation between the two low-rank approximations. The spectral norm also governs algorithmic robustness in many downstream applications, such as PCA-based learning, private clustering, and subspace tracking.

A classical spectral norm bound, derived from the Eckart–Young–Mirsky theorem [7, 16], states that $\|\tilde{A}_p - A_p\| \leq 2(\lambda_{p+1} + \|E\|)$, which holds for arbitrary matrices and noise. However, such bounds are often pessimistic and fail to exploit the structure of $A$ and $E$. More refined bounds exist in the Frobenius norm setting. For example, recent work [29, 30] shows that when $A$ is positive semidefinite and has a nontrivial eigengap $\delta_p := \lambda_p - \lambda_{p+1} \geq 4\|E\|$, and when $E$ is drawn from a complex Gaussian ensemble, one obtains: $\mathbb{E}\|\tilde{A}_p - A_p\|_F = \tilde{O}(\sqrt{p} \cdot \|E\| \cdot \frac{\lambda_p}{\delta_p})$, which improves on the earlier reconstruction-error-based bounds of [15] by a factor of $\sqrt{p}$. However, these bounds have important limitations: They hold only *in expectation* and do not yield high-probability guarantees; They often assume Gaussian noise distributions; They are not spectral norm bounds and therefore do not directly quantify the worst-case impact on the eigenspace. These limitations prompt the following open question, raised in [29, Remark 5.3]: *Can one obtain high-probability spectral norm bounds for $\|\tilde{A}_p - A_p\|$ under natural structural assumptions on $A$ and realistic noise models?*

**Our contributions.** We resolve the open question posed in [29, Remark 5.3], proving new *high-probability spectral norm bounds* for low-rank approximation under symmetric perturbations. Our results rely on natural structural assumptions on $A$ and $E$ and yield the first such guarantees for differentially private PCA (DP-PCA).

- **Two high-probability spectral norm bounds.** Under the same eigengap condition as [29], $\delta_p := \lambda_p - \lambda_{p+1} \geq 4\|E\|$, we prove $\|\tilde{A}_p - A_p\| = O\left(\|E\| \cdot \frac{\lambda_p}{\delta_p}\right)$ and $\|\tilde{A}_p - A_p\| = \tilde{O}\left(\|E\| + r^2 x \cdot \frac{\lambda_p}{\delta_p}\right)$, where $r$ is the *halving distance* (a measure of spectral decay) and $x := \max_{i,j \leq r} |u_i^\top E u_j|$ quantifies noise–eigenspace alignment (Theorems 2.1–2.2). In addition, our contour-based framework extends to a broader class of spectral functionals $f(A)$ (beyond $f(A) = A$), encompassing matrix powers, exponentials, and trigonometric transforms; see Theorem 2.3.

- **Spectral utility bounds for DP-PCA.** Our first bound yields a high-probability spectral norm utility guarantee for differentially private PCA under sub-Gaussian noise, improving existing Frobenius-norm bounds by up to a factor of $\sqrt{p}$ (Corollary 2.4). While prior work has achieved spectral norm guarantees in iterative or multi-pass settings [17, 18], our contribution concerns the *direct noise-addition* model, where this appears to be the first such result. For matrices with low stable rank and weak eigenspace–noise interaction, our second bound further improves by up to $\sqrt{n}$.

- **Novel analytical technique: contour bootstrapping.** Our proof relies on a *contour bootstrapping* argument (Lemma 3.1), which provides a new way to analyze the contour representation of perturbations [19, 26, 35], enabling analysis of a broader class of spectral functionals (Theorem 2.3). The bootstrapping argument here is a generalization of the argument used to handle eigenspaces perturbation introduced in [37].
- **Empirical validation.** We benchmark our bounds on real covariance matrices under both Gaussian and Rademacher noise. Across datasets and noise regimes, the predicted error closely matches empirical behavior and consistently surpasses classical baselines, confirming the sharpness and robustness of our theoretical results (Section 4).

## 2  Main results

**Main spectral norm bound.** For clarity, we state our main bounds assuming $A \in \mathbb{R}^{n \times n}$ is positive semi-definite (PSD); extensions to symmetric matrices appear in Section D. Let $\lambda_1 \geq \cdots \geq \lambda_n \geq 0$ be the eigenvalues of $A$, with corresponding orthonormal eigenvectors $u_1, \ldots, u_n$, and define the eigengap $\delta_k := \lambda_k - \lambda_{k+1}$. Given a real symmetric perturbation matrix $E$, we let $\tilde{A} := A + E$, and define $A_p$ and $\tilde{A}_p$ as the best rank-$p$ approximations of $A$ and $\tilde{A}$, respectively. Our goal is to bound the spectral error $\|\tilde{A}_p - A_p\|$.

**Theorem 2.1** (**Main spectral bound – PSD**). *If $4\|E\| \leq \delta_p$, then:* $\|\tilde{A}_p - A_p\| \leq O(\|E\| \cdot \frac{\lambda_p}{\delta_p})$.

The $O(\cdot)$ notation here hides a small universal constant (less than 7), which we have not optimized; see Section D.1 for the proof of the generalization to the symmetric setting, of which this theorem is a special case. For Wigner noise—i.e., a symmetric matrix $E$ with i.i.d. sub-Gaussian entries of mean 0 and variance 1—we have $\|E\| = (2 + o(1))\sqrt{n}$ with high probability [41, 43], so Theorem 2.1 reduces to $\|\tilde{A}_p - A_p\| = O\left(\sqrt{n}\,\frac{\lambda_p}{\delta_p}\right)$. The right-hand side is explicitly noise-dependent, addressing a key limitation of the classical Eckart–Young–Mirsky bound. Moreover, in many widely studied structured models (e.g., spiked covariance, stochastic block, and graph Laplacian models), one typically has $\lambda_p = O(\delta_p)$, yielding the clean bound $O(\|E\|)$. This rate is theoretically tight: for instance, when $A$ is a PSD diagonal matrix and $E = \mu I_n$ for some $\mu > 0$, we have $\|\tilde{A}_p - A_p\| = \mu = \|E\|$.

**Gap condition.** Our assumption $4\|E\| < \delta_p$ aligns with standard conditions in prior work, including [29, 30], and is satisfied in many well-studied matrix models—such as spiked covariance (Wishart) models, deformed Wigner ensembles, stochastic block models, and kernel matrices for clustering. It also arises naturally in classical perturbation theory [12, 26, 28]. Empirical analyses [29, Section B] further show that this condition holds for real-world datasets commonly used in private matrix approximation (e.g., the 1990 U.S. Census and the UCI Adult dataset [3, 11]). Hence, Theorem 2.1 operates under a mild and broadly applicable assumption, satisfied across both theoretical models and practical benchmarks.

**Comparison to the Eckart–Young–Mirsky bound.** Using $\lambda_p = \delta_p + \lambda_{p+1}$, Theorem 2.1 rewrites as $\|\tilde{A}_p - A_p\| = O(\|E\| + \lambda_{p+1} \cdot \frac{\|E\|}{\delta_p})$. This improves on the E-Y-M bound $O(\|E\| + \lambda_{p+1})$ when $\lambda_{p+1} \gg \|E\|$, by a factor of $\min\{\frac{\lambda_{p+1}}{\|E\|}, \frac{\delta_p}{\|E\|}\}$. For example, consider a matrix with spectrum $\{10n, 9n, \ldots, n, n/2, 1, \ldots, 1\}$ and $p = 10$. For Gaussian noise with $\|E\| = O(\sqrt{n})$, E-Y-M yields $O(n)$ error, while our bound gives $O(\sqrt{n})$, a $\sqrt{n}$-factor gain.

**Comparison to Mangoubi-Vishnoi bounds [29, 30].** Our bound also improves upon the Frobenius norm bounds of [29, 30], which under the same gap assumption yield: $\mathbb{E}\|\tilde{A}_p - A_p\|_F = \tilde{O}(\sqrt{p}\|E\| \cdot \frac{\lambda_p}{\delta_p})$. We eliminate the $\sqrt{p}$ factor, upgrade from expectation to high probability, and support real-valued, non-Gaussian noise models. A more detailed comparison appears later in this section (Corollary 2.4), where we analyze implications for differentially private PCA.

**Proof technique: contour bootstrapping.** Unlike prior analyses [29, 30], which rely on Dyson Brownian motion and tools from random matrix theory (see Section A, our proof of Theorem 2.1 uses a contour-integral representation of the rank-$p$ projector. This approach, which we call *contour bootstrapping*, isolates the top-$p$ eigenspace via complex-analytic techniques and avoids power-series or Davis–Kahan-type expansions. It enables tighter, structure-aware spectral bounds and ex-

tends naturally to refined perturbation results (Theorem 2.2) and general spectral functionals (Theorem 2.3). Full details appear in Section 3.

**Refined bound via eigenspace interaction.** To sharpen our analysis, we incorporate fine-grained structure of the eigenspace and its interaction with the noise. Inspired by the recent works [33, 38], we start with the observation that the rank-$p$ perturbation is primarily influenced by the cluster of eigenvalues near $\lambda_p$, and the interaction between $E$ and the corresponding eigenvectors. To control these factors, we define the *halving distance* $r$ (w.r.t the index $p$) as the smallest integer such that $\lambda_{r+1} \leq \lambda_p/2$, and *interaction term* $x := \max_{1 \leq i,j \leq r} |u_i^\top E u_j|$, measuring the alignment between the noise $E$ and the top-$r$ eigenvectors of $A$. This yields a refined spectral norm bound:

**Theorem 2.2** (**Interaction-aware bound**). *If* $4\|E\| \leq \delta_p$, *then* $\|\tilde{A}_p - A_p\| \leq \tilde{O}(\|E\| + r^2 x \cdot \frac{\lambda_p}{\delta_p})$.

See Section D.2 for the proof and its generalization to the symmetric setting. This bound improves upon the basic eigengap bound $O\left(\|E\| \cdot \frac{\lambda_p}{\delta_p}\right)$ when the interaction term $r^2 x$ is small. This occurs, for instance, when (i) $A$ has low stable rank or clustered eigenvalues (e.g., spiked models, multi-cluster Laplacians), (ii) the noise $E$ is random and approximately orthogonal to the leading eigenspace, or (iii) $\lambda_p/\delta_p$ is large but $x = \tilde{O}(1)$ and $r = \tilde{O}(1)$. In such regimes, the bound simplifies to $\tilde{O}\left(\|E\| + \frac{\lambda_p}{\delta_p}\right)$, yielding up to a $\sqrt{n}$-factor improvement over Theorem 2.1. This highlights the benefit of explicitly incorporating spectral decay and noise–eigenspace alignment when analyzing noise-robust low-rank approximations.

In practice, many public DP datasets (e.g., Census, Adult, KDD) have small dimensions and modest eigenspace decay, the simple bound is more effective. However, the refined bound becomes especially informative in large-scale or synthetically structured settings. Thus, the two bounds are best viewed as *complementary*: the first is robust and broadly applicable, while the second highlights structural regimes where stronger stability is provable.

**Extension to spectral functionals.** Beyond approximating $A$ itself, many applications involve low-rank approximations of spectral functions $f(A)$, such as $A^k$, $\exp(A)$, or $\cos(A)$; see [7, 44]. Our contour-based analysis extends naturally to this broader setting. Let $f_p(A) := \sum_{i=1}^p f(\lambda_i) u_i u_i^\top$ denote the best rank-$p$ approximation of $f(A)$. We obtain the following general perturbation bound.

**Theorem 2.3** (**Perturbation bounds for general functions**). *If* $4\|E\| \leq \delta_p$, *then*

$$\|f_p(\tilde{A}) - f_p(A)\| \leq O\left(\max_{z \in \Gamma_1} \|f(z)\| \cdot \frac{\|E\|}{\delta_p}\right),$$

*where* $\Gamma_1$ *is the rectangle with vertices* $(x_0, T), (x_1, T), (x_1, -T), (x_0, -T)$ *with*

$$x_0 := \lambda_p - \frac{\delta_p}{2}, x_1 := 2\lambda_1, T := 2\lambda_1.$$

The $O(\cdot)$ notation hides a small universal constant (less than 4), which we have not attempted to optimize; see Section F for details. For example, let $f(z) = z^3$, so that $f_p(\tilde{A})$ and $f_p(A)$ correspond to the best rank-$p$ approximations of $\tilde{A}^3$ and $A^3$, respectively. Since $\max_{z \in \Gamma_1} \|f(z)\| \leq 64\|A\|^3$, Theorem 2.3 yields $\|\tilde{A}_p^3 - A_p^3\| = O\left(\|A\|^3 \cdot \|E\|/\delta_p\right)$. This result applies to many important classes of functions—e.g., polynomials, exponentials, and trigonometric functions—and hence we expect it to be broadly useful. However, Theorem 2.3 does not apply to non-entire functions such as $f(z) = z^c$ for non-integer $c$, where singularities obstruct the contour representation (1). In particular, when $c < 0$, the expression $f_p(A)$ is no longer the best rank-$p$ approximation to $f(A)$, so the conclusion of Theorem 2.3 is not meaningful in that setting. We note that in a related work [36], the first two authors present an extension of the setting $f(z) = z^{-1}$.

**Application: differentially private low-rank approximation.** We now apply our spectral norm bound to analyze a standard differentially private (DP) mechanism for releasing a low-rank approximation of a sensitive matrix $A$, commonly assumed to be a sample covariance matrix and hence PSD. Under $(\varepsilon, \delta)$-DP [14], the Gaussian mechanism releases $\tilde{A} := A + E$, where $E$ is a symmetric matrix with i.i.d. Gaussian entries scaled to sensitivity $\Delta = O(\sqrt{\log(1/\delta)}/\varepsilon)$. A common post-processing step is to compute $\tilde{A}_p$, the best rank-$p$ approximation of $\tilde{A}$. Prior analyses [3, 15, 30] focused primarily on Frobenius norm or reconstruction error. For instance, [30] showed that under complex Wigner noise and a moderate eigengap, $\mathbb{E}\|\tilde{A}_p - A_p\|_F \leq \frac{\sqrt{pn}\lambda_p}{\delta_p}$ up to lower-order terms.

Since $\|\tilde{A}_p - A_p\| \leq \|\tilde{A}_p - A_p\|_F$, the above inequality implies an expected spectral norm error of $\tilde{O}\left(\sqrt{pn}\,\frac{\lambda_p}{\delta_p}\right)$. In contrast, our bound yields the following high-probability spectral norm guarantee:

**Corollary 2.4** (**Application to differential privacy**). *Let $A$ be PSD and $E$ be a real or complex Wigner matrix. If $\delta_p \geq 8.01\sqrt{n}$, then with probability $1 - o(1)$, $\|\tilde{A}_p - A_p\| \leq O(\sqrt{n} \cdot \frac{\lambda_p}{\delta_p})$.*

This follows directly from Theorem 2.1, using the fact that $\|E\| = O(\sqrt{n})$ with high probability for Wigner matrices [40, 43]. Compared to [30], this result provides a spectral norm (rather than Frobenius) guarantee, holds with high probability instead of in expectation, applies to both real and complex Wigner noise, removes the $\log^{\log \log n} n$ factor, and eliminates restrictive assumptions such as $\lambda_1 \leq n^{50}$. It also improves the dependence on $p$ by a factor of $\sqrt{p}$, thereby resolving the open question posed in [30, Remark 5.3].

The spectral norm better captures subspace distortion, which is critical in applications like private PCA. Unlike Frobenius or reconstruction error—both of which may remain small even when $\tilde{A}_p$ deviates significantly from the true top-$p$ eigenspace—the spectral norm reflects worst-case directional error and is thus a more reliable utility metric. This distinction is empirically validated in Figure 3. Moreover, Corollary 2.4 further yields high-probability Frobenius norm and reconstruction error bounds on the perturbation of low-rank approximations:

$$\|\tilde{A}_p - A_p\|_F \leq O\big(\sqrt{pn} \cdot \tfrac{\lambda_p}{\delta_p}\big), \text{ and } \big|\|\tilde{A}_p - A\| - \|A_p - A\|\big| \leq O\big(\sqrt{n} \cdot \tfrac{\lambda_p}{\delta_p}\big).$$

Finally, while Corollary 2.4 is stated for sub-Gaussian noise, Theorem 2.1 extends to any symmetric perturbation satisfying the norm and gap conditions, including subsampled or quantized Gaussians and Laplace noise. We leave the detailed analysis of these settings to future work.

Table 1: Summary table of perturbation bounds on $\tilde{A}_p - A_p$ for noise $E$.

|  | Bound type | Norm | Noise model | Assumption | Extra factor vs $\|E\|$ |
|---|---|---|---|---|---|
| EYM bound | High-probability | Spectral | Real and Complex | None | $O\left(1 + \frac{\lambda_{p+1}}{\|E\|}\right)$ |
| M-V bound [29] | Expectation | Frobenius | GOE (real) | $\delta_i > 4\|E\| \, \forall \, 1 \leq i \leq p$ | $O\left(\frac{\sqrt{p}\lambda_p}{\delta_p}\right)$ |
| M-V bound [30] | Expectation | Frobenius | GUE (complex) | $\delta_p > 2\|E\|, \lambda_1 < n^{50}$ | $\tilde{O}\left(\frac{\sqrt{p}\lambda_p}{\delta_p}\right)$ |
| Thm. 2.1 | High-probability | Spectral | Real and Complex | $\delta_p > 4\|E\|$ | $O\left(\frac{\lambda_p}{\delta_p}\right)$ |
| Thm. 2.2 | High-probability | Spectral | sub-Gaussian | $\delta_p > 4\|E\|, \mathrm{rank}A = \tilde{O}(1)$ | $\tilde{O}\left(1 + \frac{\lambda_p}{\delta_p\|E\|}\right)$ |

"EYM" and "M–V" denote the Eckart–Young–Mirsky and [29, 30] bounds, respectively.

**Alternative methods for approximating $A_p$.** Hardt and Price [17, 18] proposed a random iterative method which, under the condition $\delta_p \gg \sqrt{n} \log n$, produces a rank-$k$ approximation $A'$ of $A_p$ with $k = p + O(1)$, satisfying the trade-off bound $\|A' - A_p\| = \tilde{O}\left(\sqrt{n}\,\frac{\lambda_1}{\delta_p}\,\max_{1 \leq i \leq n}\|u_i\|_\infty\right)$, where $u_i$ denotes the eigenvectors of $A$.

If *at least one* eigenvector $u_i$ is localized (i.e., $\max_{1 \leq i \leq n}\|u_i\|_\infty = 1/\tilde{O}(1)$), this simplifies to $\tilde{O}\left(\sqrt{n}\,\frac{\lambda_1}{\delta_p}\right)$. In this regime, Theorem 2.1 achieves a smaller bound by a factor of $\tilde{O}(\lambda_1/\lambda_p)$—up to $\sqrt{n}$ when $\lambda_1 = \Theta(n)$ and $\lambda_p = \Theta(\sqrt{n})$. Furthermore, Theorem 2.2 provides an additional improvement by a factor of $O\left(\min\left\{\frac{\sqrt{n}}{r^2}, \frac{\lambda_1}{\delta_p}\right\}\right)$, which can reach $\sqrt{n}$ when $r = \tilde{O}(1)$ and $\delta_p = \Theta(\sqrt{n})$—a common regime in high-dimensional data.

If *all* eigenvectors $u_i$ are delocalized (i.e., $\max_{1 \leq i \leq n}\|u_i\|_\infty = \tilde{O}(1)/\sqrt{n}$), the Hardt–Price bound reduces to $\tilde{O}(\lambda_1/\delta_p)$. Theorem 2.1 achieves a comparable rate when $\sigma_1 = \Theta(n)$ and $\lambda_p = c\,\delta_p = \Theta(\sqrt{n})$, while Theorem 2.2 yields an improvement by a factor of $\lambda_1/\lambda_p$ whenever $r = \tilde{O}(1)$, i.e., when $A$ is approximately low-rank.

# 3 Proof outline

In the preceding section, we stated our main results—Theorems 2.1, 2.2, and 2.3. Here, we first sketch the key ideas behind the proof of Theorem 2.1, then adapt the same framework, with minor refinements, to derive Theorems 2.2 and 2.3.

The proof of Theorem 2.1 proceeds in three main steps. First, using the contour method, we obtain the contour-based bound of our perturbation $\|\tilde{A}_p - A_p\| \leq F(z) := \frac{1}{2\pi\mathbf{i}}\|\int_\Gamma z[(zI - \tilde{A})^{-1} - (zI - A)^{-1}]\|dz$. Here $\Gamma$ is a contour on the complex plane, isolating the $p$-leading eigenvalues of $A$ and $\tilde{A}$. This contour step captures the $A$–$E$ interaction that the Eckart–Young–Mirsky bound omits (see Appendix A). Secondly, we develop the *contour bootstrapping technique* (Lemma 3.1), which under the gap assumption $4\|E\| \leq \delta_p$, yields $F(z) \leq 2F_1(z)$ with $F_1(z) := \int_\Gamma \|z(zI - A)^{-1}E(zI - A)^{-1}\||dz|$. This technique (valid for any entire function $f$) replaces the traditional series expansions and the heavy analysis of the matrix-derivative operator (the limitation of the Mangoubi-Vishnoi approach [29, 30], Appendix A) with a computable quantity. Third, we construct a bespoke contour $\Gamma$— one specifically tailored so that the top-$p$ eigenvalues of $A$ and $\tilde{A}$ lie at prescribed distances from its sides. This precise alignment makes the integral defining $F_1(z)$ both tractable and essentially optimal, yielding a tight perturbation bound.

**Step 1: Representing $\|f_p(\tilde{A}) - f_p(A)\|$ via the classical contour method.** Let $\lambda_1 \geq \cdots \geq \lambda_n$ be the eigenvalues of $A$ with the corresponding eigenvectors $\{u_i\}_{i=1}^n$. We now present the contour method to bound matrix perturbations in the spectral norm. Let $\Gamma$ be a contour in $\mathbb{C}$ that encloses $\lambda_1, \lambda_2, \ldots, \lambda_p$ and excludes $\lambda_{p+1}, \lambda_{p+2}, \ldots, \lambda_n$. Let $f$ be any entire function and recall $f_p(A) = \sum_{i=1}^p f(\lambda_p)u_i u_i^\top$. Since $f$ is analytic on the whole plane $\mathbb{C}$, the well-known contour integral representation [19, 26, 35] gives us:

$$\tfrac{1}{2\pi\mathbf{i}} \int_\Gamma f(z)(zI - A)^{-1}dz = \sum_{i=1}^p f(\lambda_i)u_i u_i^\top = f_p(A).$$

Let $\tilde{\lambda}_1 \geq \cdots \geq \tilde{\lambda}_n$ denote the eigenvalue of $\tilde{A}$ with the corresponding eigenvectors $\tilde{u}_1, \tilde{u}_2, \ldots, \tilde{u}_n$. The construction of $\Gamma$ (presented later) and the gap assumption $4\|E\| < \delta_p$ ensure that the eigenvalues $\tilde{\lambda}_i$ for $1 \leq i \leq p$ lie inside $\Gamma$, while all $\tilde{\lambda}_j$ for $j > p$ remain outside. Then, similarly, we have $\frac{1}{2\pi\mathbf{i}} \int_\Gamma f(z)(zI - \tilde{A})^{-1}dz = \sum_{i=1}^p f(\tilde{\lambda}_i)\tilde{u}_i \tilde{u}_i^\top := f_p(\tilde{A})$. Thus, we obtain the following contour identity for the perturbation:

$$f_p(\tilde{A}) - f_p(A) = \tfrac{1}{2\pi\mathbf{i}} \int_\Gamma f(z)[(zI - \tilde{A})^{-1} - (zI - A)^{-1}]|dz|. \tag{1}$$

Now we bound the perturbation by the corresponding integral

$$\|f_p(\tilde{A}) - f_p(A)\| \leq \tfrac{1}{2\pi} \int_\Gamma \|f(z)[(zI - \tilde{A})^{-1} - (zI - A)^{-1}]\|dz =: F(f). \tag{2}$$

This inequality makes the interaction of $A$ and $E$ explicit and is widely used in functional perturbation analysis, e.g., [19, 26, 28, 32, 33, 37]. However, obtaining a sharp bound on its right-hand side remains a formidable analytical challenge.

**Step 2: Bounding $F \leq 2F_1$ via the contour bootstrapping method.** Attempts to control $F(f)$, the right-hand side of (2), often use series expansion and analytical tools. By repeatedly applying the resolvent formula, one can expand $f(z)[(zI - \tilde{A})^{-1} - (zI - A)^{-1}]$ into $\sum_{s=1}^\infty f(z)(zI - A)^{-1}[E(zI - A)^{-1}]^s$. This yields the bound:

$$F(f) \leq \sum_{s=1}^\infty F_s(f), \text{ where } F_s(f) = \tfrac{1}{2\pi} \int_\Gamma \left\|f(z)(zI - A)^{-1}[E(zI - A)^{-1}]^s\right\| |dz|.$$

One needs to estimate $F_s(f)$ for each $s$. For example, when $f(z) = 1$, [26, Part 2] bounds $F_s(1)$ by $O\left(\|E\|^s \int_\Gamma \frac{|dz|}{\min_{i\in[n]} |z-\lambda_i|^{s+1}}\right) = O\left[(\|E\|/\delta_p)^s\right]$, where $\Gamma$ is a union of vertical lines isolating $\{\lambda_i, i \in p\}$, yielding the Davis-Kahan bound $O\left(\|E\|/\delta_p\right)$. However, for $f(z) = z$ (relevant for low-rank perturbations), this approach fails as $|z| \to \infty$. These estimates are highly nontrivial and rely on deep analytical techniques, making generalization to arbitrary $f$ challenging.

Moreover, for $f(z) = 1$, under certain conditions, the dominant term is $F_1(f)$, i.e., $F(f) = O(F_1(f))$; see, e.g., [22, 27, 32, 33, 37]. In particular, using contour-bootstrapping technique, the authors in [37] proved $F(f(z) = 1) \leq 2F_1(f(z) = 1)$. Inspired by this technique, we prove that $F(f) \leq 2F_1(f)$ for any entire function $f$.

**Lemma 3.1 (Contour bootstrapping for entire function $f$).** *If $\delta_p \geq 4\|E\|$, then*

$$F(f) \leq 2F_1(f), \text{ where } F_1(f) := \tfrac{1}{2\pi} \int_\Gamma \left\| f(z)(zI - A)^{-1} E(zI - A)^{-1} \right\| |dz|.$$

Our *contour bootstrapping argument* is designed to prove Lemma 3.1. Our argument is concise and novel, avoiding the need for series expansion and convergence analysis. In the context of standard low-rank approximations, where $f(z) \equiv z$ and $f_p(A) = A_p$, we write $F(z)$ and $F_1(z)$ instead of $F(f)$ and $F_1(f)$ respectively.

**Step 3: Construction of $\Gamma$, $F_1(z)$-estimation, and proof completion of Thm. 2.1.** Given Lemma 3.1, we now need to carefully choose the contour $\Gamma$ and estimate $F_1(f)$. Constructing $\Gamma$ (so that the perturbation analysis via contour integration provides a sharp bound) is delicate; for example, the classical pick of two vertically parallel lines and any $\Gamma$ placed too near any $\lambda_i$ can blow up $F_1(z)$ to infinity. Indeed, we tailor $\Gamma$ w.r.t $F_1(z)$ as follows. First, we choose $\Gamma$ to be rectangular as this simplifies integration. To control the factor $(zI - A)^{-1}$ in the expression of $F_1(f)$, we need to ensure that the distance $|z - \lambda_i|$ for any $z \in \Gamma$ and $i \in [n]$ are relatively large. Since $\Gamma$ separates $\lambda_p$ and $\lambda_{p+1}$, this minimal distance $\min_{z \in \Gamma, i \in [n]} |z - \lambda_i|$ cannot exceed $\Theta(\delta_p)$. Thus, we simply construct $\Gamma$ through the midpoint $x_0 = \frac{\lambda_p + \lambda_{p+1}}{2}$. Finally, by setting the contour sufficiently high in the complex plane (while avoiding excessive height to prevent $|f(z)|$ from diverging), we ensure that the primary contribution to the integral is from the vertical segments of $\Gamma$. This is because the distance $|z - \lambda_i|$ is minimized on these segments. Note that, under the assumption $4\|E\| < \delta_p$, this construction ensures that the $p$-leading eigenvalues of $A$ and $\tilde{A}$ are well aligned inside the contour.

Now, in particular, to prove Theorem 2.1, we will estimate

$$2\pi F_1(z) = \int_\Gamma \left\| z(zI - A)^{-1} E(zI - A)^{-1} \right\| |dz|,$$

in which the contour $\Gamma$ is set to be a rectangle with vertices $(x_0, T), (x_1, T), (x_1, -T), (x_0, -T)$, where $x_0 := \lambda_p - \delta_p/2, x_1 := 2\lambda_1, T := 2\lambda_1$. Then, we split $\Gamma$ into four segments: $\Gamma_1 := \{(x_0, t)| -T \leq t \leq T\}$; $\Gamma_2 := \{(x, T)|x_0 \leq x \leq x_1\}$; $\Gamma_3 := \{(x_1, t)|T \geq t \geq -T\}$; $\Gamma_4 := \{(x, -T)|x_1 \geq x \geq x_0\}$.

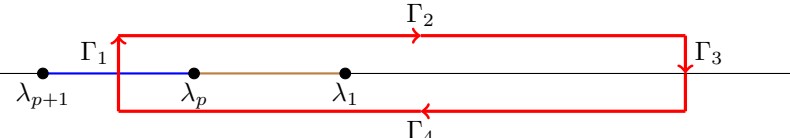

Given the construction of $\Gamma$, we have $2\pi F_1 = \sum_{k=1}^4 M_k$, where

$$M_k := \int_{\Gamma_k} \left\| z(zI - A)^{-1} E(zI - A)^{-1} \right\| |dz|.$$

Intuitively, we set $T, x_1$ large ($= 2\|A\|$) so that the main term is the integral along $\Gamma_1$, i.e., $M_1$. Indeed, factoring our $E$ and using the fact that $|z - \lambda_i| \geq |z - \lambda_p| = \sqrt{\delta_p^2 + t^2}$ for all $1 \leq i \leq n$ and $z \in \Gamma_1 := \{(x_0, t)| -T \leq t \leq T\}$, we have $M_1 \leq \int_{\Gamma_1} \|E\| \cdot \frac{|z|}{\min_{i\in[n]} |z - \lambda_i|^2} |dz| \leq \|E\| \cdot \int_{-T}^T \frac{\sqrt{x_0^2 + t^2}}{(\delta_p/2)^2 + t^2} dt$. Directly compute the integral $\int_{-T}^T \frac{\sqrt{x_0^2 + t^2}}{(\delta_p/2)^2 + t^2} dt$ (see Section E.3), we obtain:

$$M_1 \leq \|E\| \cdot O\left(x_0/\delta_p\right) = O\left(\|E\|\lambda_p/\delta_p\right).$$

By a similar manner, replace $\Gamma_1$ by $\Gamma_3 := \{(x_1, t)| -T \leq t \leq T\}$, we have

$$M_3 \leq \|E\| \cdot \int_{\Gamma_3} \frac{|z|}{\min_{i\in[n]} |z - \lambda_i|^2} |dz| \leq \|E\| \cdot \int_{\Gamma_3} \frac{\sqrt{x_1^2 + t^2}}{\lambda_1^2 + t^2} dt,$$

where the last inequality follows the fact that $\min_{i\in[n]} |z - \lambda_i| = |z - \lambda_1| = \sqrt{(x_1 - \lambda_1)^2 + t^2} = \sqrt{\lambda_1^2 + t^2}$. Directly compute the integral $\int_{-T}^T \frac{\sqrt{x_1^2 + t^2}}{\lambda_1^2 + t^2} dt$ (see Section E.3), we obtain:

$$M_3 \leq \|E\| \cdot O\left(x_1/\lambda_1\right) = O\left(\|E\|\right).$$

Similarly, $M_2, M_4 = O(\|E\|)$ (Section E.2). These estimates on $M_1, M_2, M_3, M_4$ imply $F_1(z) = O\left(\|E\| \cdot \frac{\lambda_p}{\delta_p}\right)$, which together with Lemma 3.1 proves Theorem 2.1.

**Proving the contour bootstrapping lemma (Lemma 3.1).** The first observation is that using the Sherman-Morrison-Woodbury formula $M^{-1} - (M+N)^{-1} = (M+N)^{-1}NM^{-1}$ [20] and the fact that $\tilde{A} = A + E$, we obtain

$$(zI - A)^{-1} - (zI - \tilde{A})^{-1} = (zI - A)^{-1}E(zI - \tilde{A})^{-1}.$$

Using this, we can rewrite

$$F(f) = \tfrac{1}{2\pi} \int_\Gamma \|f(z)(zI - A)^{-1}E(zI - \tilde{A})^{-1}\|\,|dz| \text{ as}$$

$$\tfrac{1}{2\pi} \int_\Gamma \|f(z)(zI - A)^{-1}E(zI - A)^{-1} - f(z)(zI - A)^{-1}E[(zI - A)^{-1} - (zI - \tilde{A})^{-1}]\|\,|dz|.$$

Using triangle inequality, we first see that $F(f)$ is at most

$$\tfrac{\int_\Gamma \|f(z)(zI-A)^{-1}E(zI-A)^{-1}\||dz|}{2\pi} + \underbrace{\tfrac{\int_\Gamma \|f(z)(zI-A)^{-1}E[(zI-A)^{-1}-(zI-\tilde{A})^{-1}]\||dz|}{2\pi}}.$$

Next is the key observation that the second term in the equation above can be rearranged and upper-bounded as follows so that the original perturbation appears again:

$$\tfrac{\max_{z\in\Gamma}\|(zI-A)^{-1}E\|}{2\pi} \int_\Gamma \|f(z)[(zI - A)^{-1} - (zI - \tilde{A})^{-1}]\|\,|dz|.$$

Thus, we have

$$F(f) \leq F_1(f) + \max_{z\in\Gamma} \|(zI - A)^{-1}E\| \cdot F(f). \tag{3}$$

Now we need our gap assumption that $4\|E\| < \delta_p$ and the construction of $\Gamma$, which imply $\min_{z\in\Gamma, i\in[n]} |z - \lambda_i| \geq \delta_p/2 \geq 2\|E\|$. Therefore, we have

$$\max_{z\in\Gamma} \|(zI - A)^{-1}E\| \leq \max_{z\in\Gamma} \|(zI - A)^{-1}\| \cdot \|E\| = \tfrac{\|E\|}{\min_{z\in\Gamma, i\in[n]} |z-\lambda_i|} \leq \tfrac{\|E\|}{2\|E\|} = \tfrac{1}{2}.$$

Together with (3), it follows that $F(f) \leq F_1(f) + \tfrac{1}{2}F(f)$. Therefore, $\tfrac{1}{2}F(f,S) \leq F_1(f,S)$. This proves Lemma 3.1.[2]

**Remark 3.2.** *Using a similar strategy, one can prove that*

$$F_1(f) \leq \max_{z\in\Gamma} \|f(z)\| \cdot \tfrac{1}{2\pi} \int_\Gamma \|(zI - A)^{-1}E(zI - A)^{-1}\||dz| \leq \max_{z\in\Gamma} \|f(z)\| \cdot \tfrac{2\|E\|}{\delta_p};$$

*see Appendix F. Together, this estimate and Lemma 3.1 prove Theorem 2.3.*

**Second upper bound of $M_1$ and proof of Theorem 2.2.** The key idea of the second bound is to replace $(zI - A)^{-1}$ by its spectral expansion $\sum_{i=1}^n \tfrac{u_i u_i^\top}{z - \lambda_i}$. Hence, $M_1$ is rewritten as $\int_{\Gamma_1} \|\sum_{1\leq i,j\leq n} \tfrac{z}{(z-\lambda_i)(z-\lambda_j)} u_i u_i^\top E u_j u_j^\top\| dz$.

There are $n^2$ terms in the expression, and the direct use of the triangle inequality cannot provide a good estimate. The next key trick is grouping up the $r$-top eigenvectors $\{u_i\}_{i=1}^r$. Formally, $M_1$ is at most

$$\int_{\Gamma_1} \|\sum_{1\leq i,j\leq r} \tfrac{z}{(z-\lambda_i)(z-\lambda_j)} u_i u_i^\top E u_j u_j^\top\||dz| + \int_{\Gamma_1} \|\sum_{n\geq i,j>r} \tfrac{z}{(z-\lambda_i)(z-\lambda_j)} u_i u_i^\top E u_j u_j^\top\||dz|$$
$$+ \int_{\Gamma_1} \|\sum_{\substack{i\leq r<j\\i>r\geq j}} \tfrac{z}{(z-\lambda_i)(z-\lambda_j)} u_i u_i^\top E u_j u_j^\top\||dz|.$$

To estimate the first term, we apply the triangle inequality. For each term, we factor out components independent of $z$ and carefully evaluate the integral. Specifically, by the triangle inequality, the first term is at most

$$\sum_{1\leq i,j\leq r} \int_{\Gamma_1} \|\tfrac{z}{(z-\lambda_i)(z-\lambda_j)} u_i u_i^\top E u_j u_j^\top\||dz| = \sum_{1\leq i,j\leq r} \int_{\Gamma_1} \tfrac{|u_i^\top E u_j|\cdot\|u_i u_j^\top\|\cdot|z|}{|(z-\lambda_i)(z-\lambda_j)|}|dz|.$$

Since $\max_{1\leq i,j\leq r} |u_i^\top E u_j| \leq x$, $\|u_i u_j^\top\| = 1$, and $\Gamma_1 := \{z \,|\, z = x_0 + \mathbf{i}t, -T \leq t \leq T\}$, the r.h.s. is at most

$$\sum_{i,j\leq r} x \int_{-T}^T \tfrac{\sqrt{x_0^2+t^2}}{\sqrt{((x_0-\lambda_i)^2+t^2)((x_0-\lambda_j)^2+t^2)}} dt \leq \sum_{i,j\leq r} x \int_{-T}^T \tfrac{|x_0|+|t|}{\sqrt{((x_0-\lambda_i)^2+t^2)((x_0-\lambda_j)^2+t^2)}} dt.$$

---

[2] The gap assumption $4\|E\| < \delta_p$ and Weyl's inequality ensure that $\tilde{\lambda}_i$ is inside the contour $\Gamma$ if and only if $1 \leq i \leq p$.

By the construction of $\Gamma_1$, we have $|x_0 - \lambda_i| \geq \frac{\delta_p}{2}$ for all $i \in [n]$. Thus, the r.h.s. is bounded by $r^2 x \int_{-T}^{T} \frac{|x_0|+|t|}{t^2+(\delta_p/2)^2} dt$, which by direct computation (see Appendix E.1 for full details) is less than or equals

$$r^2 x \left( \frac{2\pi x_0}{\delta_p} + 2\log\left(\frac{3T}{\delta_p}\right) \right) = \tilde{O}\left(r^2 x \frac{\lambda_p}{\delta_p}\right).$$

To estimate the second term, we apply matrix-norm inequalities to factor out $E$ from the integral: $\int_{\Gamma_1} \|\sum_{i,j=r}^{n} \frac{z}{(z-\lambda_i)(z-\lambda_j)} u_i u_i^\top E u_j u_j^\top \||dz| \leq \int_{\Gamma_1} |z| \cdot \|\sum_{n \geq i > r} \frac{u_i u_i^\top}{z-\lambda_i}\| \cdot \|E\| \cdot \|\sum_{n \geq i > r} \frac{u_i u_i^\top}{z-\lambda_i}\||dz|$, which is at most $\|E\| \int_{\Gamma_1} \frac{|z|}{\min_{n \geq i > r} |z-\lambda_i|^2}|dz| = \|E\| \int_{-T}^{T} \frac{\sqrt{x_0^2+t^2}}{\min_{n \geq i > r}[(x_0-\lambda_i)^2+t^2]} dt$. Moreover, by the construction of $\Gamma_1$ and the definition of $r$, $|x_0 - \lambda_i| = |(\lambda_p + \lambda_{p+1})/2 - \lambda_i| \geq |(\lambda_p + \lambda_{p+1})/2 - \lambda_{r+1}| \geq \frac{\lambda_p - \lambda_{r+1}}{2} \geq \frac{\lambda_p}{4}$, where the first inequality follows the fact $i > r$. Thus, the second term is at most

$$\|E\| \int_{-T}^{T} \frac{\sqrt{x_0^2+t^2}}{t^2+(\lambda_p/4)^2} dt \leq \tilde{O}(\|E\|);$$

see Section E.1 for the detailed estimation.

Similar to estimating the second term, the last term is also $\tilde{O}(\|E\|)$. Combining the estimates on three parts of $M_1$, we obtain $M_1 \leq \tilde{O}\left(r^2 x \frac{\lambda_p}{\delta_p} + \|E\|\right)$. Consequently, by Lemma 3.1, we finally have

$$F(z) \leq 2F_1(z) = O(M_1) = \tilde{O}\left(\|E\| + r^2 x \frac{\lambda_p}{\delta_p}\right) \quad \text{as desired.}$$

## 4 Empirical results

In this section, we empirically evaluate the sharpness of our spectral-gap bound (Theorem 2.1) in real-world settings central to privacy-preserving low-rank approximation. We compare: (1) the actual spectral error $\|\tilde{A}_p - A_p\|$, (2) our theoretical bound[3] $7\|E\| \cdot \frac{\lambda_p}{\delta_p}$, (3) and the classical Eckart–Young–Mirsky (EYM) bound $2(\|E\| + \lambda_{p+1})$. Each quantity is computed over 100 trials and 20 noise levels. Because prior bounds [15, 29, 30] apply only to Gaussian noise and involve unspecified constants, we exclude them from this evaluation.

**Setting.** We study three covariance matrices $A$ from the UCI Machine Learning Repository [13]: the 1990 US Census ($n = 69$), the 1998 KDD-Cup network-intrusion data ($n = 416$), and the Adult dataset ($n = 6$). These matrices—henceforth Census, KDD, and Adult—are standard benchmarks in DP PCA [3, 11, 29]. The low-rank parameter $p$ is chosen so that the Frobenius norm of $A_p$ contains $> 99\%$ of the Frobenius norm of $A$, giving $p = 10$ for $A = $ Census, $p = 2$ for $A = $ KDD, and $p = 4$ for $A = $ Adult [29, Section B].

Each matrix is perturbed with either GOE noise $E_1$ or Rademacher noise $E_2$, scaled by twenty evenly spaced factors ranging from 0 to 1. Note that with high probability [41, 43], $\|E_1\| = \|E_2\| = (2 + o(1))\sqrt{n}$, so the gap condition $4\|E_k\| < \delta_p$ simplifies to $8\sqrt{n} < \delta_p$. For Census ($n = 69, p = 10$), we have $\delta_p \approx 1433.99 > 8\sqrt{69} \approx 66.45$. For KDD ($n = 416, p = 2$), we get $\delta_p \approx 351.3 > 8\sqrt{416} \approx 163.2$. For Adult ($n = 6, p = 4$), we find $\delta_p \approx 37.02 > 8\sqrt{6} \approx 19.6$. Hence $4\|E_k\| < \delta_p$ holds in all tested configurations.

**Evaluation.** Each data matrix is preprocessed as follows: non-numeric entries are replaced with 0; rows shorter than the maximum length are padded with zeros; each row is scaled to unit Euclidean norm; and each column is centered to have zero mean. We compute the covariance matrix $A := M^\top M$, where $M$ is the processed data matrix. For each configuration $(A, E_k, p)$, we run 100 independent trials. In each trial, we perturb $A$ with noise $E_k \in \{E_1, E_2\}$ to form $\tilde{A} = A + E_k$, compute its best rank-$p$ approximation $\tilde{A}_p$, and measure the spectral error $\|\tilde{A}_p - A_p\|$. We compare this with our bound $7\|E_k\| \cdot \lambda_p/\delta_p$ and the classical EYM bound $2(\|E_k\| + \lambda_{p+1})$. Following standard practice, all reported values are averaged over 100 trials, with error bars shown for *Actual Error* and *Our Bound* (cap width = 3pt).

---

[3]The $O(\cdot)$ in Theorem 2.1 hides a small universal constant factor ($< 7$); see Section D.1 for details.

**Result and conclusion.** Across all experiments—the $69 \times 69$ US Census, the $416 \times 416$ KDD-Cup, and the $6 \times 6$ Adult matrix—our bound closely matches the empirical error for both Gaussian and Rademacher noise (Figs. 1–2), consistently outperforming the classical EYM estimate. (Note: the error bars for Census and KDD are too small to see.) Over all three benchmark datasets, two distinct noise models, and twenty escalation levels per model, our spectral-gap estimate never deviates from the observed error by more than a single order of magnitude. This uniform tightness, achieved without any dataset-specific tuning, demonstrates that the bound of Theorem 2.1 is not merely sufficient but practically sharp across matrix sizes spanning two orders of magnitude and privacy-motivated perturbations spanning the entire operational range. Consequently, the bound can serve as a reliable, application-agnostic error certificate for low-rank covariance approximation in both differential-privacy pipelines and more general noisy-matrix workflows.

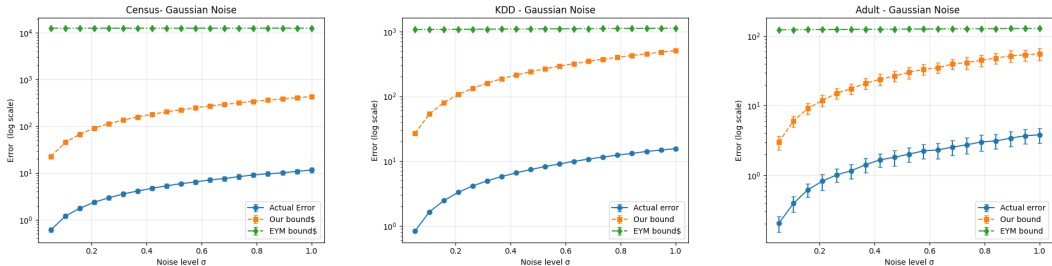

Figure 1: From Left to Right: perturbation of the Census, KDD and Adult covariance matrices by Gaussian noise. Each panel plots the actual error, our bound, and the EYM bound; error bars indicate standard deviation over 100 trials.

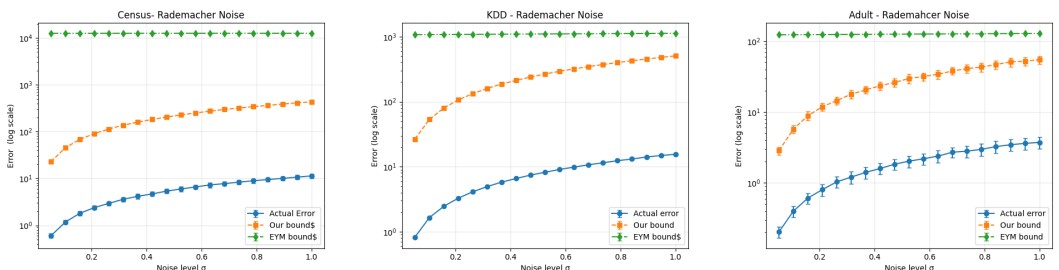

Figure 2: Low-rank approximation errors under Rademacher perturbations. From left to right: the Census, KDD and Adult covariance matrices.

## 5 Conclusion and future work

We established new spectral norm perturbation bounds for low-rank approximations that explicitly account for the interaction between a matrix $A$ and its perturbation $E$. Our results extend the Eckart—Young–Mirsky theorem, improving upon prior Frobenius-norm-based analyses. A key contribution is a novel application of the *contour bootstrapping* technique, which simplifies spectral perturbation arguments and enables refined estimates. Our bounds provide sharper guarantees for differentially private low-rank approximations with high probability spectral norm bounds that improve upon prior results. We also extended our approach to general spectral functionals, broadening its applicability.

Several limitations and open questions remain. While spectral norm error bounds are standard and widely used in both theoretical and applied settings, can we extend our analysis to other structured metrics such as Schatten-$p$ norm, the Ky Fan norm, or subspace affinity norm? Can our bounds be further refined for matrices with specific spectral structures, such as polynomial or exponential decay? What can be the threshold for the gap assumption so that one still obtains a meaningful bound beyond the Eckart–Young–Mirsky theorem?[4] Additionally, real-world noise often exhibits structured dependencies—can our techniques be adapted to handle sparse or correlated perturbations?

---

[4]For an empirical comparison between our new bound and the Eckart–Young–Mirsky bound beyond the gap condition $4\|E\| < \delta_p$, see Section C.

## Acknowledgments

This work was funded in part by NSF Award CCF-2112665, Simons Foundation Award SFI-MPS-SFM-00006506, and NSF Grant AWD 0010308.

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

# Contents

# A  Limitations of prior approaches

This section explains why existing perturbation methods fail to yield spectral norm bounds of the form $\|\tilde{A}_p - A_p\|$ that incorporate interaction between $A$ and the perturbation $E$.

**Eckart–Young–Mirsky: lack of interaction sensitivity.**  Let $\sigma_1 \geq \sigma_2 \geq \cdots \geq \sigma_n \geq 0$ denote the singular values of $A$. The Eckart–Young–Mirsky theorem gives $\|A - A_p\| = \sigma_{p+1}$, and by the triangle inequality:

$$\|\tilde{A}_p - A_p\| \leq \|A - A_p\| + \|\tilde{A} - A\| + \|\tilde{A} - \tilde{A}_p\| \leq \sigma_{p+1} + \|E\| + \tilde{\sigma}_{p+1} \leq 2(\sigma_{p+1} + \|E\|),$$

where the final step uses Weyl's inequality [46]. While this bound is assumption-free, it is uninformative in regimes where $\sigma_{p+1} \gg \|E\|$, which are common in practice. The key limitation is that the triangle inequality treats $A$ and $E$ independently, failing to capture how structure or spectral gaps in $A$ might mitigate the effect of $E$.

**Mangoubi–Vishnoi: Frobenius only, spectral norm intractable.**  The strategy of [29, 30] models noise as a continuous-time matrix-valued Brownian motion:

$$A(t) := A + tE = A + B(t),$$

with eigen-decomposition

$$A(t) = U(t) \operatorname{Diag}[\lambda_1(t), \ldots, \lambda_n(t)] \, U(t)^\top,$$

where $U(t) = [u_i(t)]$ and $\lambda_1(t) \geq \cdots \geq \lambda_n(t)$. The rank-$p$ approximation at time $t$ is

$$A_p(t) = U(t) \operatorname{Diag}[\lambda_1(t), \ldots, \lambda_p(t), 0, \ldots, 0] \, U(t)^\top.$$

The total perturbation is then expressed as an integral:

$$\tilde{A}_p - A_p = \int_0^1 dA_p(t).$$

Using properties of Dyson Brownian motion and Itô calculus, they derive a Frobenius-norm identity:

$$\mathbb{E}\left\| \int_0^1 dA_p(t) \right\|_F^2 = \sum_{i=1}^n \int_0^1 \left( \mathbb{E}\left[ \sum_{j \neq i} \frac{(\lambda_i - \lambda_j)^2}{(\lambda_i(t) - \lambda_j(t))^2} \right] + \left( \sum_{j \neq i} \frac{\lambda_i - \lambda_j}{(\lambda_i(t) - \lambda_j(t))^2} \right)^2 \right) dt.$$

Bounding these expressions depends on repulsion properties of the eigenvalues; for GOE matrices, Weyl's inequality suffices, while for GUE matrices, stronger gap estimates are used.

Although this method captures the spectral structure of $A$ and interaction with $E$, it only yields Frobenius-norm bounds. Extending it to the spectral norm would require controlling

$$\|\tilde{A}_p - A_p\| = \left\| \int_0^1 dA_p(t) \right\|,$$

which entails bounding the operator norm of the full stochastic process. This requires detailed control over the dynamics of $U(t)$ and $\lambda(t)$, including their correlations—none of which are tractable with current techniques.

Moreover, for generalized functionals such as $\|f_p(\tilde{A}) - f_p(A)\|$, the problem becomes even harder: one must analyze $\int_0^1 df_p(A(t))$, which involves matrix-valued analytic functions under random perturbation, a setting far beyond existing random matrix tools.

In contrast, our approach bypasses these limitations by using a complex-analytic representation of spectral projectors that directly captures interaction between $A$ and $E$, yielding sharp spectral norm bounds under broad assumptions.

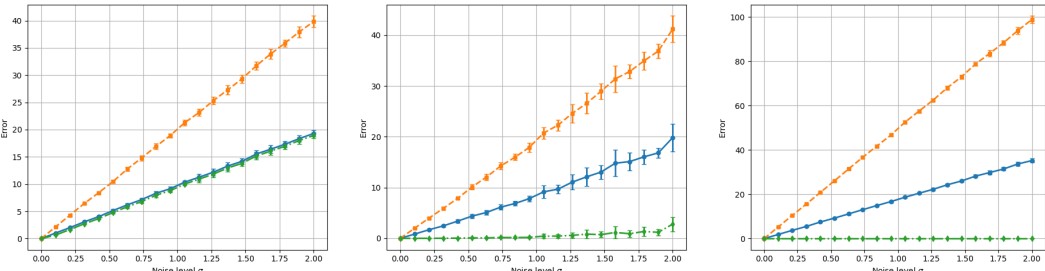

Figure 3: **Comparison of error metrics under Gaussian perturbation.** *Left:* Synthetic PSD matrix with exponentially decaying spectrum ($n = 50$, $p = 5$); *Center:* 1990 US Census covariance matrix ($n = 69$, $p = 5$); *Right:* 1998 KDD-Cup covariance matrix ($n = 416$, $p = 5$). Each plot reports the spectral norm error $\|\tilde{A}_p - A_p\|$, Frobenius norm error $\|\tilde{A}_p - A_p\|_F$, and change-in-error $\left|\|A - A_p\| - \|A - \tilde{A}_p\|\right|$, as functions of Gaussian noise level $\sigma$. Error bars reflect standard deviation over 20 trials.

## B  Comparison of error metrics

This section studies three common metrics for low-rank approximation under perturbation—namely: - the spectral-norm error $\|\tilde{A}_p - A_p\|$, - the Frobenius-norm error $\|\tilde{A}_p - A_p\|_F$, and - the "change-in-error" $\left|\|A - A_p\| - \|A - \tilde{A}_p\|\right|$.

We compare these metrics both empirically (through Monte Carlo simulations) and theoretically. Empirically, we examine how the metrics behave under Gaussian noise applied to both synthetic and real-world matrices (Figure 3). Theoretically, we analyze their interpretability and limitations, highlighting that while Frobenius norms capture aggregate error and change-in-error quantifies residual shifts, only the spectral norm controls worst-case subspace distortion.

A simple $2 \times 2$ example (Example B.1) further illustrates how residual-based measures can completely mask subspace drift, underscoring the robustness and interpretability of the spectral norm for tasks such as private low-rank approximation.

**Empirical comparison of utility metrics.**  We perform three Monte Carlo experiments under additive Gaussian perturbations. The first uses a synthetic PSD matrix $A \in \mathbb{R}^{50 \times 50}$ with exponentially decaying eigenvalues $\lambda_i = 0.8^i$, and sets $p = 5$. The second and third use real-world covariance matrices derived from: - the 1990 US Census dataset ($n = 69$), - the 1998 KDD-Cup dataset ($n = 416$).

All datasets are drawn from the UCI Machine Learning Repository [13] and have been widely used in private matrix approximation and PCA [30, 29, 11].

In each setting, we compute the best rank-$p$ approximation $A_p$, perturb $A$ with symmetric Gaussian noise of varying standard deviation $\sigma$, and measure:

1. Spectral norm deviation: $\|\tilde{A}_p - A_p\|$,
2. Frobenius norm deviation: $\|\tilde{A}_p - A_p\|_F$,
3. Change-in-error: $\left|\|A - A_p\| - \|A - \tilde{A}_p\|\right|$.

As shown in Figure 3, the Frobenius norm error grows fastest, reflecting total energy deviation. The change-in-error metric remains much smaller and, in the real-world cases, nearly flat, suggesting it may fail to capture meaningful distortion. Notably, in the synthetic case (left), the spectral norm error closely tracks the change-in-error—despite their differing intent—which may result from near-alignment of the top subspaces. However, such behavior is not guaranteed in general.

**Theoretical distinction between utility metrics.**  Frobenius norm bounds of the form $\|\tilde{A}_p - A_p\|_F \le \varepsilon_F$ aggregate squared deviations across all directions, but may hide large errors in in-

dividual components. Spectral norm bounds $\|\tilde{A}_p - A_p\| \leq \varepsilon$ directly constrain the worst-case deviation and are thus more reliable in sensitive applications such as differentially private PCA.

In contrast, residual-error metrics such as $\|A - A_p\| - \|A - \tilde{A}_p\|$ are commonly used for their analytical convenience. However, they reflect only changes in residual energy and are insensitive to subspace movement. In particular, this metric can be nearly zero even when the top-$p$ eigenspaces have shifted significantly.

Given the spectral decompositions
$$A_p = U_p \operatorname{Diag}(\lambda_1, \ldots, \lambda_p, 0, \ldots, 0) U_p^\top, \quad \tilde{A}_p = \tilde{U}_p \operatorname{Diag}(\tilde{\lambda}_1, \ldots, \tilde{\lambda}_p, 0, \ldots, 0) \tilde{U}_p^\top,$$
the change-in-error vanishes whenever $U_p U_p^\top \approx \tilde{U}_p \tilde{U}_p^\top$ and $\lambda_{p+1}$ is large. Such conditions are typical when noise $E$ is small and $p \leq \operatorname{sr}(A) := \sum_{i=1}^n \lambda_i / \lambda_1$. Moreover, standard perturbation results imply
$$\|U_p U_p^\top - \tilde{U}_p \tilde{U}_p^\top\| = \tilde{O}\left(\frac{\|E\|}{\lambda_p} + \frac{1}{\delta_p}\right) \quad [33, 38].$$

**Example B.1** (**Rank-1 rotation in** $\mathbb{R}^2$). *Let*
$$A = \begin{pmatrix} 1 & 0 \\ 0 & 0 \end{pmatrix}, \quad p = 1,$$
*so that $A_p = A$. Define the rotated matrix*
$$\tilde{A} = R_\theta A R_\theta^\top, \quad where \quad R_\theta = \begin{pmatrix} \cos\theta & -\sin\theta \\ \sin\theta & \cos\theta \end{pmatrix}.$$
*Then $\tilde{A}_p = \tilde{A}$, and although the top eigenspace has rotated by $\theta$, the change-in-error is zero:*
$$\|A - A_p\| = \|A - \tilde{A}_p\| = 0.$$
*Yet the true subspace drift is visible in:*
$$\|\tilde{A}_p - A_p\| = |\sin\theta|, \qquad \|\tilde{A}_p - A_p\|_F = \sqrt{2}|\sin\theta|.$$

This example highlights the limitations of residual-based utility metrics and illustrates why spectral norm deviation provides a more reliable and interpretable signal of approximation quality under perturbation.

In summary, both our analysis and experiments support the use of the spectral norm as the most informative and robust error metric for evaluating private low-rank approximations. Unlike Frobenius and residual metrics, it captures the worst-case directional distortion and provides a tighter connection to subspace stability.

## C Empirical evaluation beyond gap assumption

In this section, we empirically compare (1) the actual spectral error $\|\tilde{A}_p - A_p\|$, (2) our theoretical bound $7\|E\| \cdot \frac{\lambda_p}{\delta_p}$, (3) and the classical Eckart–Young–Mirsky (EYM) bound $2(\|E\| + \lambda_{p+1})$ in the setting beyond the gap assumption that $4\|E\| < \delta_p$.

**Setup.** We conducted a simulation on a covariance matrix $A$ with $n = 2000$, derived from the Alon colon-cancer microarray dataset [2]. The low-rank parameter $p$ is chosen so that the Frobenius norm of $A_p$ contains $> 95\%$ of the Frobenius norm of $A$, giving $p = 9$ with $\lambda_p \approx 46.29$. We first computed $\delta_p$. Gaussian noise was then added in the form $E = \alpha \cdot \mathcal{N}(0, I_n)$, with $\alpha$ chosen over 11 evenly spaced values such that
$$\frac{\|E\|}{\delta_p} \in \{0.05, 0.10, \ldots, 0.50\}.$$
For each $\alpha$, we computed the following quantities:

- the true error: $\|\tilde{A}_p - A_p\|$,
- the classical EYM bound: $2(\|E\| + \sigma_{p+1})$,
- our bound: $7\|E\| \cdot \frac{\lambda_p}{\delta_p}$,
- the ratios $\frac{\text{our bound}}{\text{true error}}$ and $\frac{\text{our bound}}{\text{classical bound}}$.

**Results.** Table 2 summarizes the results. The ratio $\frac{\text{our bound}}{\text{true error}}$ remains remarkably stable even beyond the regime $4\|E\| < \delta_p$ (i.e., $\frac{\|E\|}{\delta_p} < .25$), and our bound outperforms the classical bound precisely when $4\|E\| < \delta_p$ (i.e., $\frac{\|E\|}{\delta_p} < .25$).

Table 2: Comparison of bounds under increasing noise levels.

| $\|E\|/\delta_p$ | 0.05 | 0.10 | 0.15 | 0.20 | 0.25 | 0.30 | 0.35 | 0.40 | 0.45 | 0.50 |
|---|---|---|---|---|---|---|---|---|---|---|
| $\frac{\text{our bound}}{\text{true error}}$ | 90.17 | 88.27 | 87.02 | 89.83 | 89.44 | 87.81 | 88.39 | 89.29 | 87.08 | 87.26 |
| $\frac{\text{our bound}}{\text{classical bound}}$ | 0.20 | 0.40 | 0.60 | 0.79 | 0.98 | 1.17 | 1.36 | 1.53 | 1.70 | 1.88 |

# D   Extensions of Theorem 2.1 and Theorem 2.2 to the symmetric matrices

In this section, we extend Theorem 2.1 and Theorem 2.2 to the setting where $A$ is a symmetric matrix. These extensions are naturally important since the data in real-world applications is often arbitrary, making it natural for the eigenvalues of $A$ to span both signs. While singular value decomposition (SVD) could be used to apply Theorem 2.1 or Theorem 2.2, singular value gaps are typically small. By working directly with eigenvalues, we exploit the fact that the eigenvalue gap $\delta_k = \lambda_k - \lambda_{k+1}$ is significantly large when $\lambda_k \cdot \lambda_{k+1} < 0$.

## D.1   Extension of Theorem 2.1 to the symmetric matrices

To simplify the presentation, we assume that the eigenvalues (singular values) are different, so the eigenvectors (singular vectors) are well-defined (up to signs). However, our results hold for matrices with multiple eigenvalues. Let $A, E$ be $n \times n$ real symmetric matrices, and let $1 \leq p \leq n$ denote the rank of approximation. Let $\lambda_k$ be the $k$th largest eigenvalue of $A$ and $u_k$ be the corresponding orthonormal eigenvector. Let $\tilde{A} := A + E$. Let $A_p, \tilde{A}_p$ denote the best rank-$p$ approximations of $A$ and $\tilde{A}$ respectively. Define $1 \leq k \leq p$ such that the set of the top $p$ singular values corresponds to $\{\lambda_{\pi(1)}, \ldots, \lambda_{\pi(p)}\} = \{\lambda_1, \ldots, \lambda_k > 0 \geq \lambda_{n-(p-k)+1}, \ldots, \lambda_n\}$. In other words, the $p$th singular value of $A$ is either $\lambda_k$ or $|\lambda_{n-(p-k)+1}|$. Let $\delta_i := \lambda_i - \lambda_{i+1}$, for $i \in [n-1]$. Theorem 2.1 is extended to the following result.

**Theorem D.1 (Extension of Theorem 2.1 to the symmetric matrices).** *If* $4\|E\| \leq \min\{\delta_k, \delta_{n-(p-k)}\}$, *and* $2\|E\| < \sigma_p - \sigma_{p+1}$, *then*

$$\left\|\tilde{A}_p - A_p\right\| \leq 6\|E\| \left( \log\left(\frac{6\sigma_1}{\delta_k}\right) + \frac{\lambda_k}{\delta_k} + \log\left|\frac{6\sigma_1}{\delta_{n-(p-k)}}\right| + \frac{|\lambda_{n-(p-k)+1}|}{\delta_{n-(p-k)}} \right).$$

Note that when $A$ is not PSD, $\{|\tilde{\lambda}_1|, \ldots, |\tilde{\lambda}_k|, |\tilde{\lambda}_{n-(p-k)+1}|, \ldots, |\tilde{\lambda}_n|\}$ may not correspond to the $p$ leading singular values of $\tilde{A}$. This issue is resolved by enforcing the singular-value gap condition $\sigma_p - \sigma_{p+1} > 2\|E\|$. Indeed, by Weyl's inequality, given $\sigma_p - \sigma_{p+1} > 2\|E\|$, we have

$$\tilde{\lambda}_k \geq \lambda_k - \|E\| \geq \sigma_p - \|E\| = \sigma_{p+1} + \delta - \|E\|$$
$$\geq |\lambda_{n-(p-k)}| + \delta - \|E\| \geq |\tilde{\lambda}_{n-(p-k)}| + \delta - 2\|E\| > |\tilde{\lambda}_{n-(p-k)}|,$$

here $\delta = \sigma_p - \sigma_{p+1}$. By a similar argument, we also have $|\tilde{\lambda}_{n-(p-k)+1}| > \tilde{\lambda}_{k+1}$. Therefore,

$$\{\tilde{\lambda}_{\pi(1)}, \tilde{\lambda}_{\pi(2)}, \ldots, \tilde{\lambda}_{\pi(p)}\} = \{\tilde{\lambda}_1 \geq \tilde{\lambda}_2 \geq \ldots \geq \tilde{\lambda}_k > 0 \geq \tilde{\lambda}_{n-(p-k)+1} \geq \tilde{\lambda}_{n-(p-k)+2} \geq \ldots \geq \tilde{\lambda}_n\},$$

as we want. Note that the gap condition of eigenvalues cannot guarantee this fact. For example, consider the following matrices

$$A = \begin{pmatrix} 30\sqrt{n} & 0 \\ 0 & -28\sqrt{n} \end{pmatrix}, E = \begin{pmatrix} -2\sqrt{n} & 0 \\ 0 & -2\sqrt{n} \end{pmatrix}, \text{ then } \tilde{A} = \begin{pmatrix} 28\sqrt{n} & 0 \\ 0 & -30\sqrt{n} \end{pmatrix}.$$

Here, clearly, $S = \{1\}$, $\tilde{S} = \{1\}$ and $|\lambda_1|$ is the largest singular value of $A$, but $|\tilde{\lambda}_1|$ is not the largest singular value of $\tilde{A}$ ($\tilde{\lambda}_1$ is still the largest eigenvalue).

**Proof of Theorem D.1**   Let $1 \le k \le p$ be a natural number such that

$$\{\lambda_{\pi(1)}, \lambda_{\pi(2)}, \dots, \lambda_{\pi(p)}\} = \{\lambda_1, \lambda_2, \dots, \lambda_k > 0 \ge \lambda_{n-(p-k)+1}, \lambda_{n-(p-k)+2}, \dots, \lambda_n\}.$$

Thus, we can split $A_p$ as $A_k + B_{p-k}$, in which

$$B_{p-k} = \sum_{n \ge i \ge n-(p-k)+1} \lambda_i u_i u_i^\top.$$

Similarly, $\tilde{A}_p = \tilde{A}_k + \tilde{B}_{p-k}$. Therefore,

$$\left\| \tilde{A}_p - A_p \right\| = \left\| \tilde{A}_k + \tilde{B}_{p-k} - A_k - B_{p-k} \right\| \le \left\| \tilde{A}_k - A_k \right\| + \left\| \tilde{B}_{p-k} - B_{p-k} \right\|.$$

Applying the contour bootstrapping argument on $\left\| \tilde{A}_k - A_k \right\|$ with contour $\Gamma^{[1]}$ and on $\left\| \tilde{B}_{p-k} - B_{p-k} \right\|$ with another contour $\Gamma^{[2]}$ (we define these contours later), we obtain

$$
\begin{aligned}
\frac{\| \tilde{A}_k - A_k \|}{2} &\le F_1^{[1]} := \tfrac{1}{2\pi} \int_{\Gamma^{[1]}} \left\| z(zI - A)^{-1} E(zI - A)^{-1} \right\| |dz|, \\
\frac{\| \tilde{B}_{p-k} - B_{p-k} \|}{2} &\le F_1^{[2]} := \tfrac{1}{2\pi} \int_{\Gamma^{[2]}} \left\| z(zI - A)^{-1} E(zI - A)^{-1} \right\| |dz|, \\
\text{and hence,} & \\
\left\| \tilde{A}_p - A_p \right\| &\le 2 \left( F_1^{[1]} + F_1^{[2]} \right).
\end{aligned}
\tag{4}
$$

We set $\Gamma^{[1]}$ and $\Gamma^{[2]}$ to be rectangles, whose vertices are

$$\Gamma^{[1]} : (a_0, T), (a_1, T), (a_1, -T), (a_0, -T) \text{ with } a_0 := \lambda_k - \delta_k/2, a_1 := 2\sigma_1, T := 2\sigma_1;$$

and

$$\Gamma^{[2]} : (b_0, T), (b_1, T), (b_1, -T), (b_0, -T) \text{ with } b_0 := \lambda_{n-(p-k)+1} + \delta_{n-(p-k)}/2, b_1 := -2\sigma_1, T := 2\sigma_1.$$

Now, we are going to bound $F_1^{[1]}$. First, we split $\Gamma^{[1]}$ into four segments:

- $\Gamma_1 := \{(a_0, t) | -T \le t \le T\}$.
- $\Gamma_2 := \{(x, T) | a_0 \le x \le a_1\}$.
- $\Gamma_3 := \{(a_1, t) | T \ge t \ge -T\}$.
- $\Gamma_4 := \{(x, -T) | a_1 \ge x \ge a_0\}$.

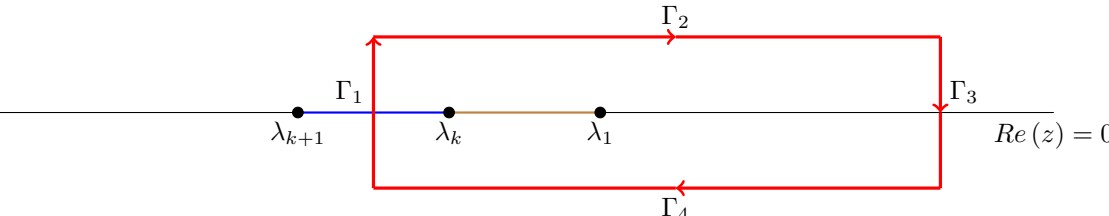

Therefore,

$$F_1^{[1]} = \sum_{l=1}^{4} \int_{\Gamma_l} \left\| z(zI - A)^{-1} E(zI - A)^{-1} \right\| |dz|.$$

Notice that

$$\left\| z(zI - A)^{-1} E(zI - A)^{-1} \right\| \le \|E\| \frac{|z|}{\min_{i \in [n]} |z - \lambda_i|^2},$$

we further obtain

$$2\pi F_1^{[1]} \le \|E\| \left( \sum_{l=1}^{4} N_l \right),$$

in which

$$N_l := \int_{\Gamma_l} \frac{|z|}{\min_i |z - \lambda_i|^2} |dz| \text{ for } l = 1, 2, 3, 4.$$

We use the following lemmas, whose proofs are delayed to the next section.

**Lemma D.2.** *Under the assumption of Theorem D.1,*

$$N_1 \leq \frac{2\pi a_0}{\delta_k} + 4\log\left|\frac{3T}{\delta_k}\right|.$$

**Lemma D.3.** *Under the assumption of Theorem D.1,*

$$N_3 \leq \frac{\pi a_1}{|a_1 - \lambda_1|} + 4\log\left|\frac{3T}{a_1 - \lambda_1}\right|.$$

**Lemma D.4.** *Under the assumption of Theorem D.1,*

$$N_2, N_4 \leq \frac{\sqrt{2}(a_1 - a_0)}{T},$$

Since $p < n$, then $k + 1 > n - (p - k) + 1$ and hence $k + 1 \notin \{\pi(1), \ldots, \pi(p)\}$. It means $|\lambda_{k+1}| \leq \lambda_k$. Thus $0 \leq a_0 \leq \lambda_k$, and hence

$$N_1 \leq \frac{2\pi\lambda_k}{\delta_k} + 4\log\left|\frac{6\sigma_1}{\delta_k}\right|.$$

By the setting that $a_1 = T = 2\sigma_1$,

$$N_2, N_4 \leq \frac{\sqrt{2}a_1}{T} = \sqrt{2},$$

$$N_3 \leq \frac{2\pi\sigma_1}{2\sigma_1 - \lambda_1} + 4\log\left|\frac{3T}{a_1 - \lambda_1}\right| \leq \frac{2\pi\sigma_1}{\sigma_1} + 4\log\left|\frac{6\sigma_1}{\sigma_1}\right| = 2\pi + 4\log 6.$$

Thus, using above estimates, we obtain

$$
\begin{aligned}
F_1^{[1]} &\leq \frac{\|E\|}{2\pi}\left(2\pi + 4\log 6 + 2\sqrt{2} + \frac{2\pi\lambda_k}{\delta_k} + 4\log\left|\frac{6\sigma_1}{\delta_k}\right|\right) \\
&\leq \frac{\|E\|}{2\pi}\left(15\log\left|\frac{6\sigma_1}{\delta_k}\right| + \frac{2\pi\lambda_k}{\delta_k}\right) \\
&\leq 3\|E\|\left(\log\left|\frac{6\sigma_1}{\delta_k}\right| + \frac{\lambda_k}{\delta_k}\right).
\end{aligned}
\tag{5}
$$

Applying a similar argument on contour $\Gamma^{[2]}$, we obtain

$$F_1^{[2]} \leq 3\|E\|\left(\log\left|\frac{6\sigma_1}{\delta_{n-(p-k)}}\right| + \frac{|\lambda_{n-(p-k)+1}|}{\delta_{n-(p-k)}}\right). \tag{6}$$

Combining (4), (5) and (6), we complete our proof.

## D.2 Extension of Theorem 2.2 to the symmetric matrices

Let $A$ be a symmetric matrix with eigenvalues $\lambda_1 \geq \lambda_2 \geq \cdots \geq \lambda_n$, in which $\lambda_n$ is not necessarily positive. Recall the setting from the previous section that $1 \leq k \leq p$ is the positive integer such that the set of the top $p$ singular values is $\{\lambda_{\pi(1)}, \ldots, \lambda_{\pi(p)}\} = \{\lambda_1, \ldots, \lambda_k > 0 \geq \lambda_{n-(p-k)+1}, \ldots, \lambda_n\}$. To extend Theorem 2.2, we first generalize the definition of the *halving distance* $r$ and *interaction term* $x$ as follows. Let $r_1, r_2$ respectively be the smallest positive integer satisfying $\frac{\lambda_k}{2} \leq \lambda_k - \lambda_{r_1+1}$, and $\frac{|\lambda_{n-(p-k)+1}|}{2} \leq \lambda_{n-r_2+1} - \lambda_{n-(p-k)+1}$. Define the "halving distance" $r := \max\{r_1, r_2\}$. Next, let $x_1 := \max_{1 \leq i,j \leq r_1} |u_i^\top E u_j|$ and $x_2 := \max_{1 \leq i,j \leq r_2} |u_{n-i+1}^\top E u_{n-j+1}|$. Define the interaction parameter $\bar{x} := \max\{x_1, x_2\}$.

**Theorem D.5 (Extension of Theorem 2.2 to the symmetric matrices).** *Assume that $4\|E\| \leq \min\{\delta_k, \delta_{n-(p-k)}\}$ and $2\|E\| < \sigma_p - \sigma_{p+1}$, then*

$$\left\|\tilde{A}_p - A_p\right\| \leq 12\left(\|E\| + r^2\bar{x}\right)\left(\log\left(\frac{6\sigma_1}{\delta_k}\right) + \log\left(\frac{6\sigma_1}{\delta_{n-(p-k)}}\right)\right) + 30r^2\bar{x}\left(\frac{\lambda_k}{\delta_k} + \frac{|\lambda_{n-(p-k)+1}|}{\delta_{n-(p-k)}}\right).$$

**Proof of Theorem D.5** First, we still split $(\tilde{A}_p, A_p)$ into $(A_k, B_{p-k}, \tilde{A}_k, \tilde{B}_{p-k})$ and apply the contour bootstrapping argument on $\left\|\tilde{A}_k - A_k\right\|, \left\|\tilde{B}_{p-k} - B_{p-k}\right\|$. We also obtain

$$\left\|\tilde{A}_p - A_p\right\| \leq 2\left(F_1^{[1]} + F_1^{[2]}\right).$$

However, we will treat $F_1^{[1]}, F_1^{[2]}$ a bit differently. Indeed,

$$2\pi F_1^{[1]} \leq M_1 + \|E\| (N_2 + N_3 + N_4),$$

in which

$$M_1 := \int_{\Gamma_1} \left\| z(zI - A)^{-1} E(zI - A)^{-1} \right\| |dz| = \int_{\Gamma_1} \left\| \sum_{1 \leq i,j \leq n} \frac{z}{(z-\lambda_i)(z-\lambda_j)} u_i u_i E u_j u_j^\top \right\| |dz|,$$

and

$$N_l := \int_{\Gamma_l} \frac{|z|}{\min_{i \in [n]} |z-\lambda_i|^2} |dz| \text{ for } l \in \{2, 3, 4\}.$$

We additionally use the following lemma (its proof will be delayed in the next section).

**Lemma D.6.** *Under the assumption of Theorem D.5,*

$$M_1 \leq r^2 \bar{x} \left( \frac{2\pi a_0}{\delta_k} + 2\log\left(\frac{6\sigma_1}{\delta_k}\right) \right) + (20 + 4\pi/\log(10)\|E\| \log\left(\frac{10\sigma_1}{\delta_k}\right).$$

Together with the estimates for $N_2, N_3, N_4$ from the previous section, we obtain

$$2\pi F_1^{[1]} \leq r^2 \bar{x} \left( \frac{2\pi\lambda_k}{\delta_k} + 2\log\left(\frac{3T}{\delta_k}\right) \right) + (20 + \frac{4\pi}{\log 10})\|E\| \log\left(\frac{5T}{\delta_k}\right) + \|E\| \left(2\sqrt{2} + 2\pi + 4\log 6\right)$$

$$\leq r^2 \bar{x} \left( \frac{2\pi\lambda_k}{\delta_k} + 2\log\left(\frac{6\sigma_1}{\delta_k}\right) \right) + (20 + \frac{4\pi}{\log 10})\|E\| \log\left(\frac{10\sigma_1}{\delta_k}\right) + \frac{2\sqrt{2}+2\pi+4\log 6}{\log 10}\|E\| \log\left(\frac{10\sigma_1}{\delta_k}\right).$$

Thus,

$$F_1^{[1]} \leq 6 \left( \|E\| \log\left(\frac{10\sigma_1}{\delta_k}\right) + r^2\bar{x}\frac{\lambda_k}{\delta_k} + r^2\bar{x} \log\left(\frac{10\sigma_1}{\delta_k}\right) \right). \tag{7}$$

Similarly,

$$F_1^{[2]} \leq 6 \left( \|E\| \log\left(\frac{10\sigma_1}{\delta_{n-(p-k)}}\right) + r^2\bar{x}\frac{|\lambda_{n-(p-k)+1}|}{\delta_{n-(p-k)}} + r^2\bar{x} \log\left(\frac{10\sigma_1}{\delta_{n-(p-k)}}\right) \right). \tag{8}$$

Therefore, combining (4), (7), and (8), we finally obtain

$$\left\| \tilde{A}_p - A_p \right\| \leq 12 \left( \|E\| \log\left(\frac{36\sigma_1^2}{\delta_k\delta_{n-(p-k)}}\right) + r^2\bar{x}\frac{\lambda_k}{\delta_k} + r^2\bar{x}\frac{|\lambda_{n-(p-k)+1}|}{\delta_{n-(p-k)}} + r^2\bar{x} \log\left(\frac{36\sigma_1^2}{\delta_k\delta_{n-(p-k)}}\right) \right).$$

# E    Estimating integrals over segments

In this section, we present in detail the integral estimations mentioned in the previous section: Lemma D.2, Lemma D.3, Lemma D.6 (integration over vertical segments); and Lemma D.4 (integration over horizontal segments) . We first present a technical lemma, which is used several times in the upcoming sections.

**Lemma E.1.** *Let $a, T$ be positive numbers such that $a \leq T$. Then,*

$$\int_{-T}^{T} \frac{1}{t^2+a^2} dt \leq \frac{\pi}{a}.$$

**Proof of Lemma E.1**    We have

$$\int_{-T}^{T} \frac{1}{t^2+a^2} dt = 2\int_0^T \frac{1}{t^2+a^2}$$
$$= \frac{2}{a}\arctan(T/a)$$
$$\leq \frac{2}{a} \cdot \frac{\pi}{2} = \frac{\pi}{a}.$$

### E.1    Estimating integrals over vertical segments for interaction-dependent bound

In this Section, we now estimate $M_1$ - integral over the left vertical segment (prove Lemma D.6) and estimate $N_3$- the integral over the right vertical segment (prove Lemma D.3). First, we estimate $M_1$ as follows.

Using the spectral decomposition $(zI - A)^{-1} = \sum_{i=1}^{n} \frac{u_i u_i^\top}{(z-\lambda_i)}$, we can rewrite $M_1$ as

$$M_1 = \int_{\Gamma_1} \left\| \sum_{n \geq i,j \geq 1} \frac{z}{(z-\lambda_i)(z-\lambda_j)} u_i u_i^\top E u_j u_j^\top \right\| |dz|.$$

Define $x_1 := \max_{1 \le i, j \le r_1} |u_i^\top E u_j|$. By the triangle inequality, $M_1$ is at most

$$\int_{\Gamma_1} \left\| \sum_{1 \le i,j \le r_1} \frac{z}{(z-\lambda_i)(z-\lambda_j)} u_i u_i^\top E u_j u_j^\top \right\| |dz| + \int_{\Gamma_1} \left\| \sum_{n \ge i, j > r_1} \frac{z}{(z-\lambda_i)(z-\lambda_j)} u_i u_i^\top E u_j u_j^\top \right\| |dz|$$

$$+ \int_{\Gamma_1} \left\| \sum_{\substack{i \le r_1 < j \\ i > r_1 \ge j}} \frac{z}{(z-\lambda_i)(z-\lambda_j)} u_i u_i^\top E u_j u_j^\top \right\| |dz|.$$

Consider the first term, by the triangle inequality, we have

$$\int_{\Gamma_1} \left\| \sum_{1 \le i,j \le r_1} \frac{z}{(z-\lambda_i)(z-\lambda_j)} u_i u_i^\top E u_j u_j^\top \right\| |dz| \le \sum_{1 \le i,j \le r_1} \int_{\Gamma_1} \left\| \frac{z}{(z-\lambda_i)(z-\lambda_j)} u_i u_i^\top E u_j u_j^\top \right\| |dz|$$

$$= \sum_{1 \le i,j \le r_1} \int_{\Gamma_1} \frac{|u_i^\top E u_j| \cdot \|u_i u_j^\top\| \cdot |z|}{|(z-\lambda_i)(z-\lambda_j)|} |dz|$$

$$\le \sum_{i,j \le r_1} x_1 \int_{-T}^{T} \frac{\sqrt{a_0^2+t^2}}{\sqrt{((a_0-\lambda_i)^2+t^2)((a_0-\lambda_j)^2+t^2)}} dt$$

$$\text{(since } \max_{1 \le i,j \le r_1} |u_i^\top E u_j| \le x_1, \|u_i u_j^\top\| = 1,$$
$$\text{and } \Gamma_1 := \{z \,|\, z = a_0 + \mathbf{i}t, -T \le t \le T\})$$

$$\le \sum_{i,j \le r_1} x_1 \int_{-T}^{T} \frac{|a_0|+|t|}{\sqrt{((a_0-\lambda_i)^2+t^2)((a_0-\lambda_j)^2+t^2)}} dt.$$

By the construction of $\Gamma_1$, we have

$$|a_0 - \lambda_i| \ge \frac{\delta_k}{2} \quad \text{for all } 1 \le i \le n. \tag{9}$$

Thus, the r.h.s. is at most

$$r_1^2 x_1 \int_{-T}^{T} \frac{|a_0|+|t|}{t^2+(\delta_k/2)^2} dt = r_1^2 x_1 \left( \int_{-T}^{T} \frac{|a_0|}{t^2+(\delta_k/2)^2} dt + \int_0^T \frac{2t}{t^2+(\delta_k/2)^2} dt \right). \tag{10}$$

By Lemma E.1, we have

$$\int_{-T}^{T} \frac{|a_0|}{t^2+(\delta_k/2)^2} dt \le \frac{2\pi|a_0|}{\delta_k} = \frac{2\pi a_0}{\delta_k} \text{(since } a_0 \ge 0).$$

The second integral is estimated by what follows.

$$\int_0^T \frac{2t}{t^2+(\delta_k/2)^2} dt = \int_{(\delta_k/2)^2}^{T^2+(\delta_k/2)^2} \frac{1}{u} du \quad (u = t^2 + (\delta_k/2)^2)$$

$$= \log\left( \frac{T^2+(\delta_k/2)^2}{(\delta_k/2)^2} \right) \tag{11}$$

$$= \log\left( \frac{4T^2+\delta_k^2}{\delta_k^2} \right) \le 2\log\left( \frac{3T}{\delta_k} \right).$$

Therefore,

$$\int_{\Gamma_1} \left\| \sum_{1 \le i,j \le r_1} \frac{z}{(z-\lambda_i)(z-\lambda_j)} u_i u_i^\top E u_j u_j^\top \right\| |dz| \le r_1^2 x_1 \left( \frac{2\pi a_0}{\delta_k} + 2\log\left( \frac{3T}{\delta_k} \right) \right). \tag{12}$$

Next, we bound the second term as follows

$$\int_{\Gamma_1} \left\| \sum_{n \ge i, j > r_1} \frac{z}{(z-\lambda_i)(z-\lambda_j)} u_i u_i^\top E u_j u_j^\top \right\| |dz|$$

$$= \int_{\Gamma_1} \left\| z \left( \sum_{n \ge i > r_1} \frac{u_i u_i^\top}{z-\lambda_i} \right) E \left( \sum_{n \ge i > r_1} \frac{u_i u_i^\top}{z-\lambda_i} \right) \right\| |dz|$$

$$\le \int_{\Gamma_1} |z| \cdot \left\| \sum_{n \ge i > r} \frac{u_i u_i^\top}{z-\lambda_i} \right\| \times \|E\| \times \left\| \sum_{n \ge i > r} \frac{u_i u_i^\top}{z-\lambda_i} \right\| |dz|$$

$$\le \|E\| \int_{\Gamma_1} \frac{|z|}{\min_{n \ge i > r_1} |z-\lambda_i|^2} |dz|$$

$$= \|E\| \int_{-T}^{T} \frac{\sqrt{a_0^2+t^2}}{\min_{n \ge i > r_1} [(a_0-\lambda_i)^2+t^2]} dt.$$

Moreover, by the construction of $\Gamma_1$ and the definition of $r_1$,

$$|a_0 - \lambda_i| = \left| \frac{\lambda_k+\lambda_{k+1}}{2} - \lambda_i \right| \ge \left| \frac{\lambda_k+\lambda_{k+1}-2\lambda_{r+1}}{2} \right| \ge \frac{\lambda_k-\lambda_{r+1}}{2} \ge \frac{\lambda_k}{4}, \tag{13}$$

where the second inequality follows the fact $i > r_1$. Thus, the r.h.s. is at most

$$\|E\| \int_{-T}^{T} \frac{\sqrt{a_0^2+t^2}}{t^2+(\lambda_k/4)^2} dt \le \|E\| \int_{-T}^{T} \frac{a_0+|t|}{t^2+(\lambda_k/4)^2} dt.$$

Similar to (10) and (11), we also have

$$
\begin{aligned}
\int_{-T}^{T} \frac{a_0+|t|}{t^2+(\lambda_k/4)^2} dt &\le \frac{4\pi a_0}{\lambda_k} + \log\left(\frac{T^2+(\lambda_k/4)^2}{(\lambda_k/4)^2}\right) \\
&\le \frac{4\pi a_0}{\lambda_k} + \log\left(\frac{2T^2}{\delta_k^2}\right) \\
&\le 4\pi + 2\log\left(\frac{2T}{\delta_k}\right) \quad \text{(since } a_0 \le \lambda_k\text{)}.
\end{aligned}
\tag{14}
$$

It follows that

$$\int_{\Gamma_1} \left\| \sum_{n \ge i,j > r_1} \frac{z}{(z-\lambda_i)(z-\lambda_j)} u_i u_i^\top E u_j u_j^\top \right\| |dz| \le \|E\| \left(4\pi + 2\log\left(\frac{2T}{\delta_k}\right)\right). \tag{15}$$

Now we consider the last term:

$$\int_{\Gamma_1} \left\| \sum_{\substack{i \le r_1 < j \\ i > r_1 \ge j}} \frac{z}{(z-\lambda_i)(z-\lambda_j)} u_i u_i^\top E u_j u_j^\top \right\| |dz| \le 2\|E\| \int_{\Gamma_1} \frac{|z|}{\min_{i \le r_1 < j} |(z-\lambda_i)(z-\lambda_j)|} |dz|.$$

By (13) and (9), the r.h.s. is at most

$$
\begin{aligned}
2\|E\| \int_{-T}^{T} \frac{|z|}{\sqrt{(t^2+(\delta_k/2)^2)(t^2+(a_0-\lambda_{r+1})^2)}} dt &= 4\|E\| \int_{0}^{T} \frac{\sqrt{a_0^2+t^2}}{\sqrt{(t^2+(\delta_k/2)^2)(t^2+(a_0-\lambda_{r+1})^2)}} dt \\
&\le 4\|E\| \int_{0}^{T} \frac{a_0+t}{\sqrt{(t^2+(\delta_k/2)^2)(t^2+(a_0-\lambda_{r+1})^2)}} dt.
\end{aligned}
\tag{16}
$$

Moreover, $\int_{0}^{T} \frac{a_0+t}{\sqrt{(t^2+(\delta_k/2)^2)(t^2+(a_0-\lambda_{r+1})^2)}} dt$ equals

$$
\begin{aligned}
&\int_{0}^{T} \frac{a_0 dt}{\sqrt{(t^2+(\delta_k/2)^2)(t^2+(a_0-\lambda_{r+1})^2)}} + \int_{0}^{T} \frac{t dt}{\sqrt{(t^2+(\delta_k/2)^2)(t^2+(a_0-\lambda_{r+1})^2)}} dt \\
&= \int_{0}^{T} \frac{a_0 dt}{\sqrt{(t^2+(\delta_k/2)^2)(t^2+(a_0-\lambda_{r+1})^2)}} + \frac{1}{2}\log\left(\frac{(T^2+(\delta_k/2)^2)(a_0-\lambda_{r+1})^2}{(T^2+(a_0-\lambda_{r+1})^2)\delta_k/2}\right) \\
&\le \frac{a_0}{\max\{\delta_k/2, a_0-\lambda_{r+1}\}} \times \log\left(\frac{T+\sqrt{T^2+\min\{\delta_k/2, a_0-\lambda_{r+1}\}^2}}{\min\{\delta_k/2, a_0-\lambda_{r+1}\}}\right) + \frac{1}{2}\log\left(\frac{(T^2+(\delta_k/2)^2)(a_0-\lambda_{r+1})}{(T^2+(a_0-\lambda_{r+1})^2)\delta_k/2}\right).
\end{aligned}
$$

Note that $a_0 - \lambda_{r+1} \ge \delta_k/2$ and $a_0 - \lambda_{r+1} + \delta_k/2 = \lambda_k - \lambda_{r+1} \ge \frac{\lambda_k}{2}$. Therefore, $a_0 - \lambda_{r+1} = \max\{\delta_k/2, a_0 - \lambda_{r+1}\} \ge \frac{\lambda_k}{4}$. We further obtain that $\int_{0}^{T} \frac{a_0+t}{\sqrt{(t^2+(\delta_k/2)^2)(t^2+(a_0-\lambda_{r+1})^2)}} dt$ is at most

$$\frac{a_0}{\lambda_k/4} \cdot \log\left(\frac{5T}{\delta_k}\right) + \frac{1}{2}\log\left(\frac{2T}{\delta_k}\right) \le 4.5 \log\left(\frac{5T}{\delta_k}\right). \tag{17}$$

The estimates (16) and (17) together imply that the last term is at most

$$18\|E\| \left(\frac{5T}{\delta_k}\right). \tag{18}$$

Combining (12), (15) and (18), we finally obtain that $M_1$ is at most

$$
\begin{aligned}
&r_1^2 x_1 \left(\frac{2\pi a_0}{\delta_k} + 2\log\left(\frac{3T}{\delta_k}\right)\right) + \|E\| \left(4\pi + 2\log\left(\frac{2T}{\delta_k}\right)\right) + 18\|E\|\log\left(\frac{5T}{\delta_k}\right) \\
&\le r_1^2 x_1 \left(\frac{2\pi a_0}{\delta_k} + 2\log\left(\frac{6\sigma_1}{\delta_k}\right)\right) + (20 + 4\pi/\log(10)\|E\|\log\left(\frac{10\sigma_1}{\delta_k}\right) \quad \text{(since } \log\left(\frac{10\sigma_1}{\delta_k}\right) \ge \log 10\text{)} \\
&\le r^2 \bar{x} \left(\frac{2\pi a_0}{\delta_k} + 2\log\left(\frac{6\sigma_1}{\delta_k}\right)\right) + (20 + 4\pi/\log(10)\|E\|\log\left(\frac{10\sigma_1}{\delta_k}\right).
\end{aligned}
\tag{19}
$$

This proves Lemma D.6.

Next, we estimate $N_3$. Notice that

$$
\begin{aligned}
N_3 &= \int_{\Gamma_3} \frac{|z|}{\min_i |z-\lambda_i|^2} |dz| \\
&= \int_{-T}^{T} \frac{\sqrt{a_1^2+t^2}}{\min_{i\in[n]}[(a_1-\lambda_i)^2+t^2]} dt \ \ (\text{since } \Gamma_3 := \{z \,|\, z = a_1 + \mathbf{i}t, -T \le t \le T\}) \\
&\le \int_{-T}^{T} \frac{\sqrt{a_1^2+t^2}}{t^2+(a_1-\lambda_1)^2} dt \\
&\le \int_{-T}^{T} \frac{|a_1|}{t^2+(a_1-\lambda_1)^2} dt + \int_{-T}^{T} \frac{|t|}{t^2+(a_1-\lambda_1)^2} dt \\
&\le \left| \frac{\pi a_1}{a_1-\lambda_1} \right| + 2\log\left( \left| \frac{T}{a_1-\lambda_1} \right|^2 + 1 \right) \ \ (\text{by Lemma E.1}) \\
&\le \frac{\pi a_1}{a_1-\lambda_1} + 4\log\left| \frac{3T}{a_1-\lambda_1} \right|.
\end{aligned}
$$

This proves Lemma D.3.

### E.2 Estimating integrals over horizontal segments

We are going to bound $N_2, N_4$ - integral over top horizontal segment (prove Lemma D.4). We have

$$
\begin{aligned}
N_2 &= \int_{\Gamma_2} \frac{|z|}{\min_{i\in[n]} |z-\lambda_i|^2} |dz| \\
&= \int_{a_0}^{a_1} \frac{\sqrt{x^2+T^2}}{\min_{i\in[n]}((x-\lambda_i)^2+T^2)} dx \ \ (\text{since } \Gamma_2 := \{z \,|\, z = x + \mathbf{i}T, a_0 \le x \le a_1\}) \\
&\le \int_{a_0}^{a_1} \frac{\sqrt{2}T}{T^2} dx \ \ (\text{since } x \le a_1 \le T) \\
&= \frac{\sqrt{2}|a_1-a_0|}{T}.
\end{aligned}
$$

By similar arguments, we also obtain

$$
N_4 \le \frac{\sqrt{2}|a_1-a_0|}{T}.
$$

These estimates on $N_2, N_4$ prove Lemma D.4.

### E.3 Estimating integrals over vertical segments for non-interaction bound

In this Section, we estimate $N_1$, proving Lemma D.2. The estimation of $N_3$ follows the case of the interaction-dependent bound at the end of Section E.1.

$$
\begin{aligned}
N_1 &= \int_{\Gamma_1} \frac{|z|}{\min_i |z-\lambda_i|^2} |dz| \\
&= \int_{-T}^{T} \frac{\sqrt{a_0^2+t^2}}{\min_{i\in[n]}[(a_0-\lambda_i)^2+t^2]} dt \ \ (\text{since } \Gamma_1 := \{z \,|\, z = a_0 + \mathbf{i}t, -T \le t \le T\}) \\
&\le \int_{-T}^{T} \frac{\sqrt{a_0^2+t^2}}{t^2+(\delta_k/2)^2} dt \ \ (\text{by (9)}) \\
&\le \int_{-T}^{T} \frac{|a_0|}{t^2+(\delta_k/2)^2} dt + \int_{-T}^{T} \frac{|t|}{t^2+(\delta_k/2)^2} dt \\
&\le \left| \frac{2\pi a_0}{\delta_k} \right| + 2\log\left( \left| \frac{2T}{\delta_k} \right|^2 + 1 \right) \ \ (\text{by Lemma E.1}) \\
&\le \left| \frac{2\pi a_0}{\delta_k} \right| + 4\log\left| \frac{3T}{\delta_k} \right|.
\end{aligned}
$$

This proves Lemma D.2.

## F Perturbation of matrix functionals - Theorem 2.3

In this section, we complete the delayed proof of Theorem 2.3. By Remark 3.2, to prove Theorem 2.3, we need to show that

$$
2\pi F_1(1) := \int_{\Gamma} \|(zI-A)^{-1}E(zI-A)^{-1}\| |dz| = 4\pi \frac{\|E\|}{\delta_p},
$$

in which the contour $\Gamma$ is set to be a rectangle with vertices

$$(x_0, T), (x_1, T), (x_1, -T), (x_0, -T), \text{ where } x_0 := \lambda_p - \delta_p/2, x_1 := 2\lambda_1, T := 2\lambda_1.$$

We split $\Gamma$ into four segments:

$$\Gamma_1 := \{(x_0, t) | -T \leq t \leq T\}; \Gamma_2 := \{(x, T) | x_0 \leq x \leq x_1\};$$

$$\Gamma_3 := \{(x_1, t) | T \geq t \geq -T\}; \Gamma_4 := \{(x, -T) | x_1 \geq x \geq x_0\}.$$

Therefore,

$$\int_\Gamma \|(zI - A)^{-1}E(zI - A)^{-1}\||dz| = \sum_{i=1}^{4} M_k, \tag{20}$$

in which

$$M_i := \int_{\Gamma_i} \|(zI - A)^{-1}E(zI - A)^{-1}\||dz| \text{ for } i \in \{1, 2, 3, 4\}.$$

By a similar strategy from previous section, we bound $M_1$ as follows. Notice that

$$\left\|(z - A)^{-1}E(z - A)^{-1}\right\| \leq \frac{\|E\|}{\min_{i \in [n]} |z - \lambda_i|^2}.$$

Therefore, $M_1$ is at most

$$\|E\| \cdot \int_{\Gamma_1} \frac{1}{\min_{i \in [n]} |z - \lambda_i|^2} |dz|$$

$$\leq \|E\| \cdot \int_{\Gamma_1} \frac{1}{|z - \lambda_p|^2} |dz| \text{ (since } \lambda_p \text{ is closest to } \Gamma_1 \text{ among all eigenvalues of } A)$$

$$= \|E\| \cdot \int_{-T}^{T} \frac{1}{(\delta_p/2)^2 + t^2} dt \text{ (by definition } \Gamma_1 := \{(x_0, t) | -T \leq t \leq T\} \text{ and } |x_0 - \lambda_p| = \delta_p/2)$$

$$\leq \frac{2\pi\|E\|}{\delta_p} \text{ (by Lemma E.1)}.$$

Next, we bound $M_3$ as what follows.

$$M_3 \leq \|E\| \int_{\Gamma_3} \frac{1}{\min_{i \in [n]} |z - \lambda_i|^2} |dz|$$

$$= \|E\| \int_{-T}^{T} \frac{1}{\min_{i \in [n]} ((x_1 - \lambda_i)^2 + t^2)} dt \text{ ( since } \Gamma_3 := \{z \mid z = x_1 + \mathbf{i}t, -T \leq t \leq T)$$

$$= \|E\| \int_{-T}^{T} \frac{1}{t^2 + (x_1 - \lambda_1)^2} dt$$

$$\leq \frac{\pi\|E\|}{|x_1 - \lambda_1|} \text{ (by Lemma E.1)}$$

$$= \frac{\pi\|E\|}{\lambda_1} \text{ (since } x_1 = 2\lambda_1).$$

Next, we estimate $M_2$ as

$$M_2 \leq \int_{\Gamma_2} \frac{1}{\min_{i \in [n]} |z - \lambda_i|^2} \|E\||dz| = \|E\| \int_{\Gamma_2} \frac{1}{\min_{i \in [n]} |z - \lambda_i|^2} |dz|.$$

Moreover, since $\Gamma_2 := \{z \mid z = x + \mathbf{i}T, x_0 \leq x \leq x_1\}$,

$$\int_{\Gamma_2} \frac{1}{\min_{i \in [n]} |z - \lambda_i|^2} |dz| = \int_{x_0}^{x_1} \frac{1}{\min_{i \in [n]} ((x - \lambda_i)^2 + T^2)} dx \leq \int_{x_0}^{x_1} \frac{1}{T^2} dx = \frac{|x_1 - x_0|}{T^2}.$$

Therefore, $M_2 \leq \frac{\|E\| \cdot |x_1 - x_0|}{T^2} \leq \frac{\|E\|\lambda_1}{4\lambda_1^2} = \frac{\|E\|}{4\lambda_1}$. Similarly, we also obtain that $M_4 = \frac{\|E\|}{4\lambda_1}$. These estimates on $M_1, M_2, M_3, M_4$ and Equation 20 imply

$$\int_\Gamma \|(zI - A)^{-1}E(zI - A)^{-1}\||dz| = \frac{2\pi\|E\|}{\delta_p} + (\pi + \frac{1}{4} + \frac{1}{4})\frac{\|E\|}{\lambda_1} \leq 4\pi\frac{\|E\|}{\delta_p}.$$

The last inequality follows the trivial fact that $\lambda_1 > \delta_p$ for any PSD matrix $A$. We complete the proof.

# G  Some classical perturbation bounds

This section recalls standard results referenced in Section 2, Section 3, and Section A.

**Theorem G.1 (Eckart–Young–Mirsky bound [16]).** *Let $A, \tilde{A} \in \mathbb{R}^{n \times n}$, and let $A_p$, $\tilde{A}_p$ denote their respective best rank-$p$ approximations. Set $E := \tilde{A} - A$. Then,*

$$\|\tilde{A}_p - A_p\| \leq 2\left(\sigma_{p+1} + \|E\|\right),$$

*where $\sigma_{p+1}$ is the $(p+1)$st singular value of $A$.*

**Theorem G.2 (Weyl's inequality [46]).** *Let $A, E \in \mathbb{R}^{n \times n}$ be symmetric, and define $\tilde{A} := A + E$. Then, for any $1 \leq i \leq n$,*

$$|\tilde{\lambda}_i - \lambda_i| \leq \|E\| \quad and \quad |\tilde{\sigma}_i - \sigma_i| \leq \|E\|,$$

*where $\lambda_i, \tilde{\lambda}_i$ are the $i$th eigenvalues of $A$ and $\tilde{A}$, and $\sigma_i, \tilde{\sigma}_i$ are the corresponding singular values.*

# H  Notation

This section summarizes the key notations used throughout the paper. Let $A, E$ be symmetric $n \times n$ matrices, and define the perturbed matrix $\tilde{A} := A + E$. Let $f$ be an entire function, and let $s \in \mathbb{N}$.

Table 3: Summary of notation

| Symbol | Definition |
| --- | --- |
| $n$ | Dimension of $A$, $\tilde{A}$ |
| $p$ | Target rank parameter |
| $A_p$ | Best rank-$p$ approximation of $A$ |
| $\tilde{A}_p$ | Best rank-$p$ approximation of $\tilde{A}$ |
| $\lambda_1 \geq \cdots \geq \lambda_n$ | Eigenvalues of $A$ in descending order |
| $\tilde{\lambda}_1 \geq \cdots \geq \tilde{\lambda}_n$ | Eigenvalues of $\tilde{A}$ in descending order |
| $\sigma_1 \geq \cdots \geq \sigma_n$ | Singular values of $A$ in descending order |
| $\delta_i$ for $i \in [n-1]$ | $i$th eigengap: $\delta_i := \lambda_i - \lambda_{i+1}$ |
| $u_i$ for $i \in [n]$ | Eigenvector of $A$ corresponding to $\lambda_i$ |
| $\tilde{u}_i$ for $i \in [n]$ | Eigenvector of $\tilde{A}$ corresponding to $\tilde{\lambda}_i$ |
| $\mathrm{sr}(A)$ | Stable rank: $\mathrm{sr}(A) := \|A\|_F^2 / \|A\|^2$ (p. 22) |
| Halving distance $r$ | Smallest integer such that $\lambda_p/2 \geq \lambda_{r+1}$ (p. 3, Thm. 2.2) |
| Interaction term $x$ | $x := \max_{1 \leq i,j \leq r} |u_i^\top E u_j|$ (p. 3, Thm. 2.2) |
| $f_p(A)$ | $f_p(A) := \sum_{i=1}^{p} f(\lambda_i) u_i u_i^\top$ (p. 4, Thm. 2.3) |
| $f_p(\tilde{A})$ | $f_p(\tilde{A}) := \sum_{i=1}^{p} f(\tilde{\lambda}_i) \tilde{u}_i \tilde{u}_i^\top$ (p. 4, Thm. 2.3) |
| $\Gamma$ | Contour enclosing $\{\lambda_1, \ldots, \lambda_p\}$ (p. 5) |
| $F(f)$ | $\frac{1}{2\pi} \int_\Gamma \|f(z)[(zI - \tilde{A})^{-1} - (zI - A)^{-1}]\| \, |dz|$ (p. 5, Eq. (2)) |
| $F_s(f)$ | $\frac{1}{2\pi} \int_\Gamma \|f(z)(zI - A)^{-1}[E(zI - A)^{-1}]^s\| \, |dz|$ (p. 6) |
| $F_1(f)$ | $\frac{1}{2\pi} \int_\Gamma \|f(z)(zI - A)^{-1} E(zI - A)^{-1}\| \, |dz|$ (p. 6, Lem. 3.1) |
| $F(z)$ | $\frac{1}{2\pi} \int_\Gamma \|z[(zI - \tilde{A})^{-1} - (zI - A)^{-1}]\| \, |dz|$ (p. 6) |
| $F_1(z)$ | $\frac{1}{2\pi} \int_\Gamma \|z(zI - A)^{-1} E(zI - A)^{-1}\| \, |dz|$ (p. 6) |
| $\| \cdot \|$ | Spectral norm |
| $\| \cdot \|_F$ | Frobenius norm |
| EYM bound | Eckart–Young–Mirsky bound |
| M–V bound | Mangoubi–Vishnoi bound |
| PSD | Positive semi-definite |

