# OpenReview forum: "Spectral Perturbation Bounds for Low-Rank Approximation with Applications to Privacy"
_NeurIPS.cc/2025/Conference — NeurIPS 2025 oral_

### Official Review · Reviewer_swEV · 2025-06-27

**Clarity:** 4
**Significance:** 4
**Originality:** 4
**Rating:** 6
**Confidence:** 3

**Summary:**

This paper studies how an additive noise matrix affects low-rank approximation. In particular, given a symmetric $n$-by-$n$ matrix $A$ and a symmetric noise matrix $E$, the goal is to bound $\|(A+E)_p - A_p\|$ for some norm $\|\cdot\|$. (Given a symmetric matrix $M$, let $M_p$ denote its best rank-$p$ approximation.)
Previous works have studied error metrics such as the Frobenius norm $\|(A+E)_p - A_p\|_F$ and the change in reconstruction error $| \|(A+E)_p - A\| - \|A_p - A\| |$.
This paper studies the spectral norm error $\|(A+E)_p - A_p\|_2$, which is more suitable in certain cases. This is validated both theoretically and in experiments in the paper.
The main result of this paper is, assuming $A$ is PSD, if $\|E\| \le (\lambda_p - \lambda_{p+1}) / 4$, then $\| (A+E)_p - A_p \| \lesssim \| E \| \cdot \frac{\lambda_p}{\lambda_p - \lambda_{p+1}}$, where $\lambda_1 \ge \cdots \ge \lambda_n \ge 0$ are the eigenvalues of $A$. (They also extend the result to general symmetric matrices. The bound is less clean than the PSD case.)
Their result is better than the Eckart–Young–Mirsky bound than a factor of $\min\{\frac{\lambda_{p+1}}{\|E\|}, \frac{\lambda_{p} - \lambda_{p+1}}{\|E\|}\}$, which is a meaningful improvement when $\lambda_{p+1} \gg \|E\|$ and $\lambda_{p} - \lambda_{p+1} \gg \|E\|$.
When translating the previous Frobenius norm bound by Mangoubi and Vishnoi to the spectral norm bound, it will be worse than the error bound in this paper by a factor of $\sqrt{p}$. Moreover, the previous Frobenius norm bound by Mangoubi and Vishnoi only holds in expectation, while this paper obtains high probability bounds.
One important application of their main result is differentially private PCA.
This paper also gets two more results: a more fine-grained bound that takes into account a measure of spectral decay of $A$ and the alignment between the noise $E$ and the top eigenvectors of $A$; a perturbation bound for general matrix functions.
The authors prove these results by using a novel contour bootstrapping method from complex analysis and extends it to a broad class of spectral functionals, including polynomials and matrix exponentials.

**Questions:**

Why consider $\|(A+E)_p - A_p\|$ instead of $\|(A+E)_p - A\|$? It seems to me a more natural goal is to approximate $A$ instead of $A_p$. (Of course, $\|(A+E)_p - A\| \le \|(A+E)_p - A_p\| + \|A_p - A\|$.)

**Ethical Concerns:**

["NO or VERY MINOR ethics concerns only"]

**Final Justification:**

The authors have addressed my questions and I keep my positive rating.

**Quality:**

4

**Strengths And Weaknesses:**

I think the nice results obtained in this paper would be interesting at least for the filed of differential privacy and matrix perturbation theory. The spectral norm error bound is not only of interest itself, but also is more preferable in some applications. This paper is also very well-written. It motivates the spectral norm error bound well, presents a clear proof outline, and gives a good explanation of why previous techniques fail.

---

> ### Author Rebuttal · Authors · 2025-07-30
>
> Thank you for your valuable comments and questions. We are glad that you find our new spectral norm bound and the contour bootstrapping technique both interesting and valuable for theoretical studies and applications. We answer your specific question below.
> ### Questions:
> > Why consider $\\|(A+E)_p -A_p\\|$  instead of $\\|(A+E)_p -A\\|$ ? It seems to me a more natural goal is to approximate A  instead of $A_p$ . (Of course, $\\|(A+E)_p - A\\| < \\|(A+E)_p -A_p\\| + \\|A_p -A\\|$ .)
>
>
> Thank you for this interesting question. We agree that in some contexts one might aim to approximate $A$ directly. However, in the setting of low-rank approximation, it is standard to treat $A_p$—the best rank-$p$ approximation to $A$—as the target.
>
> This is because any rank-$p$ matrix, including $(A + E)_p$, must incur an irreducible error of $\\|A - A_p\\|$ when approximating $A$. That component is inherent to the low-rank constraint and cannot be improved. The quantity $\\| (A+E)_p - A_p \\|$ captures the *additional* error introduced by the perturbation $E$—and thus represents the part we can meaningfully analyze and minimize.
>
> Moreover, as you noted, the triangle inequality gives:
> $$
> \\| (A+E)_p - A \\| \leq \\| (A+E)_p - A_p \\| + \\| A_p - A \\|,
> $$
> so bounding $\\| (A+E)_p - A_p \\|$ also gives insight into how well $(A+E)_p$ approximates $A$ overall.
>
> We are happy to discuss this point further in the final version if clarification would be helpful.

---

> > ### Comment · Reviewer_swEV · 2025-08-05
> >
> > Thank you for answering my question. I keep my score.

---

### Official Review · Reviewer_oLme · 2025-06-28

**Clarity:** 3
**Significance:** 4
**Originality:** 4
**Rating:** 5
**Confidence:** 3

**Summary:**

The paper presents the error bound (in terms of the spectral norm) for the rank-$p$ approximation of a symmetric matrix perturbed with a symmetric noise. The paper is written nicely with a clear motivation and description. The mathematical tools used to solve this hard problem namely contour bootstrapping are interesting in their own right and the authors seem to have found a novel application for this tool in this work. The simulation results also seem to highlight the utility of the derived bound when compared to classical baselines.

**Questions:**

Major comments:
1. It is important to clarify the following: How different is the derived bound from that obtained by applying norm equivalence to the Frobenius error bound in https://arxiv.org/abs/2211.06418? Particularly, the $\sqrt{p}$ improvement, as outlined in lines 119-123,  strongly suggests this.
2. The results in Figures 1-2 should also consider the aforementioned bound from norm equivalence for comparison.
3. Given that the result is presented with DP as the primary application, in light of the second weakness stated in the strength and weakness section, it is desirable to have an illustrative example showing where the condition $\Vert E \Vert \leq \delta_p/4$ would fail and where it will hold (especially with the increasing dimension $n$ for different values of DP parameters, $(\epsilon,\delta)$) along with an illustration on how $\lambda_p$ and $\lambda_{p+1}$ behave in such conditions.

Minor comments:
1. It has to be clearly indicated when the asymptotic bounds are probabilistic (e.g.,  $O_{\texttt{p}}$ and $O_{\texttt{p}}$) - please see https://arxiv.org/abs/2103.08721 and https://arxiv.org/abs/1108.3924.
2. Provide a reference (e.g., "High-Dimensional Statistics: A Non-Asymptotic View-point" by M. J. Wainwright) for lines 151-154.

**Ethical Concerns:**

["NO or VERY MINOR ethics concerns only"]

**Final Justification:**

The authors have addressed all my comments/queries well. They have given a convincing reasons for the points which cannot be addressed easily. Further they have also provided an example in response to my question 3 which seems correct. Since I have already given a rating of 5, I continue to retain the rating and commend the authors on the excellent work and rebuttal

**Limitations:**

yes

**Quality:**

3

**Strengths And Weaknesses:**

Strengths:
1. The derived bound is tighter than the EYM bound and leverages the symmetry of data and noise matrices. Also, the presented bounds are high probability asymptotic bounds, which are preferable over the bounds on expectation.
2. The paper adopts a new analytical approach, contour bootstrapping.

Weaknesses:
1. The paper highlights the importance of the spectral norm error bounds over the Frobenius norm error or reconstruction error. However, the spectral norm still does not present a complete picture. In my opinion, the more appropriate metric for the low-rank approximation problem would be the affinity between true and estimated subspaces (as defined in https://arxiv.org/abs/1301.2603) or, say, the Ky Fan norm.
2. The technical assumption in line 66 for the bound to hold is too restrictive in DP since DP mechanisms usually add excessive noise that also scales with the dimension

---

> ### Author Rebuttal · Authors · 2025-07-30
>
> Thank you for your valuable comments and suggestions. We are glad that you appreciate our new high-probability bound and the novel contour bootstrapping technique. We answer your specific comments and suggestions below.
> ### Comments:
> >The paper highlights the importance of the spectral norm error bounds over the Frobenius norm error or reconstruction error. However, the spectral norm still does not present a complete picture. In my opinion, the more appropriate metric for the low-rank approximation problem would be the affinity between true and estimated subspaces (as defined in https://arxiv.org/abs/1301.2603) or, say, the Ky Fan norm.
>
> Thank you for the intriguing comment and for pointing us to this interesting work.
>
> While spectral norm error bounds are standard and widely used in both theoretical and applied settings, we agree that metrics such as subspace affinity and the Ky Fan norm offer complementary insights—particularly when evaluating subspace structure or alignment.
>
> Extending our framework to these metrics would require new technical ideas, especially to control perturbations of projection operators or spectral sums. We view this as an important direction for future work and will mention this in the final version.
>
>
>
> >The technical assumption in line 66 for the bound to hold is too restrictive in DP since DP mechanisms usually add excessive noise that also scales with the dimension.
>
> Thank you for the comment. We acknowledge that the condition $4\\|E\\| < \delta_p$ may be restrictive in some settings, and we believe future work may extend our results beyond this threshold—especially for matrices with favorable spectral structure.
>
> That said, this type of gap condition has been widely adopted in the DP literature. For example, it appears in both theoretical and empirical work, including Dwork et al. (STOC 2014, Main Result 2) and Mangoubi–Vishnoi (JACM 2025, Remark 5.3). Mangoubi–Vishnoi (NeurIPS 2022, Appendix J) also provide empirical evidence that the condition holds on standard datasets such as Census and Adult—commonly used benchmarks in DP.
>
> We will revise our presentation to clarify the applicability and limitations of this assumption.
>
>
>
>
> > It is important to clarify the following: How different is the derived bound from that obtained by applying norm equivalence to the Frobenius error bound in https://arxiv.org/abs/2211.06418? Particularly, the \sqrt{p}- improvement, as outlined in lines 119-123, strongly suggests this.
>
> Thank you for the helpful comment and the opportunity to clarify.
>
> Since the matrix $\tilde{A}_p - A_p$ has rank at most $2p$, we have the standard inequality:
> $$
> \\|\tilde{A}_p - A_p\\|_F \leq \sqrt{2p} \cdot \\|\tilde{A}_p - A_p\\|.
> $$
> Thus, our spectral norm bound (Theorem 2.1) directly implies the Frobenius-norm bound in Mangoubi-Vishnoi [arXiv:2211.06418]:
> $$
> \\|\tilde{A}_p - A_p\\|_F \leq O\left( \sqrt{p} \cdot \\|E\\| \cdot \frac{\lambda_p}{\delta_p} \right)
> $$
> with high probability.
>
> However, the reverse direction does not hold in general: one only has
> $$
> \\|\tilde{A}_p - A_p\\| \leq \\|\tilde{A}_p - A_p\\|_F,
> $$
> so Frobenius-norm bounds alone do not yield spectral norm bounds. In particular, the result in [arXiv:2211.06418] only establishes an *expected* spectral norm bound of
> $$
> \mathbb{E} \\|\tilde{A}_p - A_p\\| \leq \tilde{O}\left( \sqrt{p} \cdot \\|E\\| \cdot \frac{\lambda_p}{\delta_p} \right),
> $$
> whereas our result gives a high-probability spectral norm bound of
> $$
> \\|\tilde{A}_p - A_p\\| \leq O\left( \\|E\\| \cdot \frac{\lambda_p}{\delta_p} \right).
> $$
> We will add this clarification to the final version of the paper.
>
>
>
>
> >The results in Figures 1-2 should also consider the aforementioned bound from norm equivalence for comparison.
>
> Thank you for the valuable suggestion. Bounds derived via norm equivalence—such as Theorem 9 from Dwork et al. (STOC 2014) for reconstruction error and the Frobenius-norm bounds from Mangoubi–Vishnoi (JACM 2025)—are expressed using $O$ or $\tilde{O}$ notation with unspecified constants. This makes it difficult to include them meaningfully in quantitative plots without making arbitrary choices.
>
> We will add a remark explaining this in the final version.
>
>
>
>
> > Given that the result is presented with DP as the primary application, in light of the second weakness stated in the strength and weakness section, it is desirable to have an illustrative example showing where the condition $||E||< \delta_p/4$  would fail and where it will hold (especially with the increasing dimension n  for different values of DP parameters, $(\varepsilon, \delta)$ ) along with an illustration on how $\lambda_p$  and $\lambda_{p+1}$  behave in such conditions.
>
> Thank you for the thoughtful suggestion. We agree that including an example illustrating when the assumptions fail or hold would strengthen the paper, and we have indeed kept this threshold in mind throughout.
>
> The phenomenon that $\\|\tilde{A}_p - A_p\\|$ can blow up when $\\|E\\| > \delta_p$ has been observed frequently. For a concrete example, consider PSD matrices $A$ and $E$ with spectral decompositions:
> - $A = \sum_{i=1}^n \sigma_i u_i u_i^\top$,
> - $E = \sum_{i \neq p+1} \varepsilon u_i u_i^\top + \mu u_{p+1} u_{p+1}^\top$,
>
> where $\sigma_1 > \sigma_2 > \cdots > \sigma_n > 0$, $\varepsilon > 0$ is small (e.g., $\varepsilon < \delta_i$ for all $1 \leq i \leq p$), and $\mu > \sigma_p - \sigma_{p+1}$.
> In this case, we have:
> $$
> \\|\tilde A_p - A_p\\| = \\| (\mu + \sigma_{p+1}) u_{p+1} u_{p+1}^\top - \sigma_p u_p u_p^\top \\| > \sigma_p.
> $$
>
> In the specific context of $(\epsilon, \delta)$-differential privacy, where $E \sim \mathcal{N}(0, \sigma^2 I_n)$ with $\sigma = \frac{\sqrt{\log(1/\delta)}}{\epsilon}$, we can set, for instance:
> - $\sigma_p - \sigma_{p+1} = 4\sqrt{n}$,
> - $\delta = 1/10$, $\epsilon = 1/2$.
>
> Then, with high probability, $\\|E\\| > \delta_p / 4$, and the above construction implies $\\|\tilde{A}_p - A_p\\| > \sigma_p$.
>
> We will include this example in the final version to illustrate the importance of the spectral gap condition and its relevance to practical DP settings.
>
>
>
>
> >It has to be clearly indicated when the asymptotic bounds are probabilistic (e.g., O_p and O_p ) - please see https://arxiv.org/abs/2103.08721 and https://arxiv.org/abs/1108.3924.
>
> Thank you for the comment and suggestion. We will carefully review our use of $O$ and $\tilde{O}$ throughout the paper and correct any inconsistencies. We will also include a brief clarification of our notation to distinguish high-probability bounds where appropriate.
>
> > Provide a reference (e.g., "High-Dimensional Statistics: A Non-Asymptotic View-point" by M. J. Wainwright) for lines 151-154.
>
> Thank you for pointing us to this reference. We will include a citation to Wainwright’s book in the final version.

---

> > ### Comment · Reviewer_oLme · 2025-08-03
> >
> > The authors have addressed all the comments I have raised very well and I have no more questions/comments

---

### Official Review · Reviewer_HKv5 · 2025-07-02

**Clarity:** 4
**Significance:** 4
**Originality:** 4
**Rating:** 6
**Confidence:** 4

**Summary:**

The paper studies rank-$k$ approximation under spectral norm. They give really great results and something that was definitely lacking in previous works.

**Questions:**

I have one quick question: can we use the techniques in this paper to also give high probability bound in COLT 2023 paper where the authors study the Frobenius norm metric? In my opinion, that was the weak point of the paper. I do not see that using Dyson Brownian motion based method can be used to get a high probability bound (having extensively thought on this problem since the NeurIPS 2022 paper came out).

**Ethical Concerns:**

["NO or VERY MINOR ethics concerns only"]

**Limitations:**

None.

**Paper Formatting Concerns:**

None.

**Quality:**

4

**Strengths And Weaknesses:**

Beautiful paper and amazing result!

I do not see any weakness in the paper. The writing is good. I am ready to champion the paper and do not have much to say. I would have accepted the paper just with one set of results (the bound on spectral norm error). I have thought about this problem for over three years, so I understand the limitations of previous approaches used when it comes to bound the spectral norm metric and get the high probability bound (instead of expected bound in the complex Dyson Brownian based analysis).

The weakness, if I can say, is that I was not able to read the proof, but I am looking forward to completely understanding the paper and engaging with the authors during the rebuttal phase with any questions I will have.

---

> ### Author Rebuttal · Authors · 2025-07-30
>
> Thank you for your valuable comments and suggestions. We are glad that you appreciate our results and are interested in a deeper understanding of our work. We answer your specific question below.
> ### Questions:
> > Can we use the techniques in this paper to also give high probability bound in COLT 2023 paper where the authors study the Frobenius norm metric? In my opinion, that was the weak point of the paper.
>
> Thank you for the question. Yes, our techniques can be used to obtain a high-probability bound in Frobenius norm.
>
> The most direct approach is to use the inequality
> $$
> \\|\tilde{A}_p - A_p\\|_F \leq \sqrt{2p} \cdot \\|\tilde{A}_p - A_p\\|,
> $$
> since $\tilde{A}_p - A_p$ has rank at most $2p$. Applying our spectral norm bound (Theorem 2.1 or Corollary 2.4), we obtain:
> $$
> \\|\tilde{A}_p - A_p\\|_F \leq O \left( \sqrt{p} \cdot \\|E\\| \cdot \frac{\lambda_p}{\delta_p} \right)
> = O\left( \sqrt{p n} \cdot \frac{\lambda_p}{\delta_p} \right)
> $$
> with high probability.
>
> Alternatively, one could revisit the contour bootstrapping argument and estimate $F_1$ directly in Frobenius norm. This may require additional technical refinements to preserve sharpness, but the framework extends naturally.
>
> We will expand our discussion to include a comparison with the bounds presented in the COLT 2023 paper to clarify our improvements.

---

### Official Review · Reviewer_aeeH · 2025-07-02

**Clarity:** 3
**Significance:** 3
**Originality:** 2
**Rating:** 5
**Confidence:** 4

**Summary:**

The work provides new perturbation bounds for low rank approximations in the spectral norm, improving over classical results that follow from the Eckart-Young-Mirsky theorem.

The main feature of the bounds is that they are sensitive to the misalignment between the perturbation and the top singular vectors. This is important for applications in differential privacy, where the perturbation is typically Gaussian noise, and hence has small projection in the direction of any fixed eigenvector. The authors discuss the applications of their work to differentially private PCA.

**Questions:**

1. How do your results for DP-PCA compare to those you can achieve with the power method or other iterative methods?
2. Do your proof techniques give results in the overparameterized regime, where we try to capture the top p eigenvectors with k > p dimensions?
3. It would be interesting to see additional experiments on a larger datasets and in cases where the assumptions may not hold exactly.

**Ethical Concerns:**

["NO or VERY MINOR ethics concerns only"]

**Final Justification:**

The authors have adequately answered my first question about the comparison with the noisy power method. The discussion convinced me me that the bounds are often significantly stronger than would follow from the noisy power method for the low-rank approximation objective. This gave me a better appreciation for the significance of the results, which is why I raised my score.

**Limitations:**

yes

**Quality:**

4

**Strengths And Weaknesses:**

The paper is well written and provides helpful discussion throughout. The main results are elegant and natural, providing interesting extensions to classical results from numerical analysis and matrix theory. The proof technique sounds intriguing. Apparently, the authors have a new technique that avoids some of the classical tools like Davis-Kahan expansions. This sounds great, although I did not have time to dig into the details.

Using the new perturbation bounds, the authors give some improvements for one particular way of achieving differentially private PCA: Add noise to the original matrix and compute a low rank approximation on top of the noisy matrix. Whereas most previous results applied only to Frobenius norm, the new results apply to spectral norm. I agree with the authors that this is the much more meaningful guarantee in realistic settings.

Note, though, that this is just one way of achieving differentially private PCA. It's a well studied problem and there are numerous ways of doing it.

The authors claim that they give the first high probability bounds for differentially private PCA in the spectral norm. This is, however, not true. High probability bounds in the spectral norm already follow from the work by Hardt and Price (NeurIPS 2015) and Hardt and Roth (STOC 2013). Thos papers analyze noisy subspace iteration. The bounds are sensitive to the misalignment between the noise (gaussian) and the spectrum of the matrix. In addition, they give results for overparameterization: approximating rank p by rank k > p matrices. It's worth discussing these classical results. In which ways do the new tools give improvements over these results?

In particular, in the empirical results it would be good to compare to these methods.

---

> ### Author Rebuttal · Authors · 2025-07-30
>
> Thank you for your valuable comments and suggestions. We are glad that you appreciate our main results as natural and elegant improvements over classical bounds, and that you find our new contour bootstrapping technique of interest. We address your specific questions and concerns below.
>
> ### Questions:
> > How do your results for DP-PCA compare to those you can achieve with the power method or other iterative methods?
>
>
> Thank you for this important question. We appreciate the opportunity to clarify the comparison.
>
> Since Hardt and Price (NeurIPS 2015) is a refined and improved version of Hardt and Roth (STOC 2013), we focus on Theorem 1.4 and Corollary 4.3 from Hardt and Price.
>
> Both their work and ours aim to construct an $(\varepsilon, \delta)$-differentially private matrix that approximates $A_p$. Their approach, the Noisy Power Method (NPM), produces a rank-$k$ approximation $A'$ with $k = p + \Omega(p)$, serving as a private surrogate for $A_p$. Under the assumption $\sigma_p - \sigma_{p+1} > C \sqrt{n} \log n$ (for some constant $C$), they show that with high probability:
>
> $$
> \\|A' - A_p\\| = \tilde{O} \left( \frac{\sqrt{n \log(1/\delta)}}{\varepsilon} \cdot \frac{\sigma_1}{\delta_p} \cdot \max_{1 \leq i \leq n} \\|u_i\\|_\infty \right).
> $$
>
> In contrast, our algorithm directly perturbs $A$ and then computes a rank-$p$ approximation $\tilde{A}_p$. We obtain the following high-probability bounds:
>
> - By Theorem 2.1 and Corollary 2.4:
>   $$
>   \\|\tilde{A}_p - A_p\\| = O \left( \frac{\sqrt{n \log(1/\delta)}}{\varepsilon} \cdot \frac{\sigma_p}{\delta_p} \right).
>   $$
>
> - By Theorem 2.2:
>   $$
>   \\|\tilde{A}_p - A_p\\| = \tilde{O} \left( \frac{\sqrt{n \log(1/\delta)}}{\varepsilon} + \frac{r^2 \sigma_p}{\delta_p} \right),
>   $$
>   where $r \geq p$ is the smallest index such that $\sigma_r \leq \sigma_p / 2$.
>
> These bounds highlight two performance regimes:
>
> - Our Theorem 2.1 and Corollary 2.4 offer improved trade-offs when at least one eigenvector $u_i$ is localized. In such cases, our gain factor can be up to $\tilde{O}(\sigma_1 / \sigma_p)$ compared to NPM.
>
> - Our Theorem 2.2 is advantageous when $r^2 \ll \tilde O ( \sqrt{n} \cdot \max_{i}  \\|u_i\\|_\infty )$.
>
> This occurs, for instance, when $A$ has low stable rank—i.e., $r = \tilde O(1)$—since $\max_{i} \\|u_i \\|_\infty \geq 1/\sqrt{n}$.
>
> We will include this comparison in the final version to clarify the trade-offs between the two approaches.
>
>
>
>
> >[Related comment] The authors claim that they give the first high probability bounds for differentially private PCA in the spectral norm. This is, however, not true. High probability bounds in the spectral norm already follow from the work by Hardt and Price (NeurIPS 2015) and Hardt and Roth (STOC 2013).
>
> As noted above, prior work such as Hardt and Price (NeurIPS 2015) provides high-probability spectral norm bounds for iterative methods. We thank the reviewer for pointing this out and apologize for the oversight.
>
> Our contribution focuses on a different setting—adding noise directly to the data matrix—where, to our knowledge, our result gives the first such bound. We will qualify our claims accordingly and include proper citations and discussion in the final version.
>
>
>
>
> > Do your proof techniques give results in the overparameterized regime, where we try to capture the top $p$ eigenvectors with $k > p$ dimensions?
>
> Thank you for the interesting question. Our proof techniques do not directly yield bounds on $\\|\tilde{A}_k - A_p\\|$ for $k > p$.
>
> While it is possible to obtain a bound via the triangle inequality,
> $$\\|\tilde{A}_k - A_p\\| \leq \\|\tilde{A}_k - \tilde{A}_p\\| + \\|\tilde{A}_p - A_p\\|,
> $$
> this is not directly controlled by our main results. The limitation arises from the contour-integral argument, which is tailored to bounding $\tilde{A}_p - A_p$ using a contour $\Gamma$ that encloses only the eigenvalues $\lambda_i$ and $\tilde{\lambda}_i$ for $1 \leq i \leq p$.
>
> We view extending our framework to handle overparameterization as a promising direction for future work.
>
>  ### Other comments:
>
>
>  >It would be interesting to see additional experiments on a larger datasets and in cases where the assumptions may not hold exactly.
>
> Thank you for the helpful suggestion. We agree that including examples and simulations illustrating behavior when the assumptions do not hold could strengthen the paper.
>
> Indeed, it is well known—and we have also observed—that the error $\\|\tilde{A}_p - A_p\\|$ may blow up when $\\|E\\| > \delta_p$. For instance, consider PSD matrices $A$ and $E$ with the following spectral decompositions:
> - $A = \sum_{i=1}^n \sigma_i u_i u_i^\top$,
> - $E = \sum_{i \neq p+1} \varepsilon u_i u_i^\top + \mu u_{p+1} u_{p+1}^\top$,
>
> where $\varepsilon > 0$ is a small constant and $\mu > \sigma_p - \sigma_{p+1}$. In this case,
> $$
> \\|\tilde A_p - A_p\\| = \left\\| (\mu + \sigma_{p+1}) u_{p+1} u_{p+1}^\top - \sigma_p u_p u_p^\top \right\\| > \sigma_p.
> $$
>
> From an empirical perspective, we also conducted a simulation on a larger dataset. We used a covariance matrix $A$ (with $n=2000$) derived from the Alon colon-cancer microarray dataset. We selected $p = 9$ such that $A_p$ captures 95% of $A$ in Frobenius norm, and $\lambda_p \approx 46.29$.
>
> The simulation setup:
> - Compute $\delta_p$.
> - Add Gaussian noise $E = \alpha \cdot \mathcal{N}(0, I_n)$, where $\alpha$ ranges over 11 evenly spaced values such that $\\|E\\| / \delta_p \in \{0.05, 0.1, \ldots, 0.5\}$.
> - For each $\alpha$, compute:
>   - true error: $\|\tilde{A}_p - A_p\|$,
>   - classical bound: $2(\\|E\\| + \sigma_{p+1})$,
>   - our bound: $7\\|E\\| \cdot \frac{\lambda_p}{\delta_p}$,
>   - and the ratios:
>     - $\frac{\text{our bound}}{\text{true error}}$,
>     - $\frac{\text{our bound}}{\text{classical bound}}$.
>
> The results are summarized below:
>
> | $\|E\|/\delta_p$                          | 0.05  | 0.10  | 0.15  | 0.20  | 0.25  | 0.30  | 0.35  | 0.40  | 0.45  | 0.50  |
> |:-----------------------------------------|------:|------:|------:|------:|------:|------:|------:|------:|------:|------:|
> | $\frac{\text{our bound}}{\text{true error}}$     | 90.17 | 88.27 | 87.02 | 89.83 | 89.44 | 87.81 | 88.39 | 89.29 | 87.08 | 87.26 |
> | $\frac{\text{our bound}}{\text{classical bound}}$| 0.20  | 0.40  | 0.60  | 0.79  | 0.98  | 1.17  | 1.36  | 1.53  | 1.70  | 1.88  |
>
> These results are particularly interesting:
> - The ratio $\frac{\text{our bound}}{\text{true error}}$ remains stable even beyond the regime where $4\\|E\\| < \delta_p$.
> - Our bound outperforms the classical bound exactly when $4\\|E\\| < \delta_p$.
>
> We will consider including this example and simulation in the final version of the paper.

---

> > ### Comment · Reviewer_aeeH · 2025-08-05
> >
> > The factor \sigma_1 that you put in the error bound for the noisy power method (NPM) doesn't seem quite right to me. So, I looked it up. I'm looking at Theorem 1.3 in arXiv v4 of Hardt and Price. Note that what they call $\sigma$ is not the top singular value $\sigma_1$, but rather an expression defined in Figure 3, that's roughly $\sqrt{pL}/\epsilon$, where $L\approx\sigma_k/\delta_k$.
> >
> > So, it seems like NPM is generally better if $\max_i \|u_i\|_\infty$ is relatively small. But even if this is $1$ (as large as it could be), the current analysis only makes marginal improvements, if I'm not mistaken.
> >
> > Specifically, Theorem 1.3 in NPM gives an error like (ignoring O-tilde, epsilon, delta):
> >
> > $$
> > \frac{\sqrt{n}\sqrt{p}\sqrt{\sigma_k}}
> > {\delta_k^{1.5}}
> > $$
> >
> > Whereas Theorem 2.1 here gives:
> >
> > $$
> > \frac{\sqrt{n}\sigma_p}
> > {\delta_p}
> > $$
> >
> > These bounds are incomparable in general, but it seems that in typical settings, the NPM bound is marginally better.
> >
> > Let's assume we have a sharp eigenvalue decay so that $\delta_k\approx c\sigma_k$ and $\delta_p\approx c'\sigma_p$ for some constants $c, c'$. In this case, your bound ends up being mostly $\sqrt{n}.$ The NPM bound is better so long as $\delta_k\ge 1$ and $\sigma_k\ge \sqrt{p}.$ Assuming the matrix is 0/1 and $p$ is constant, this is the typical setting.
> >
> > To summarize, for incoherent matrices (singular vector misaligned with the standard basis), the NPM bound is better. For localized singular vectors, the bounds are very similar. The NPM bound has the advantage that it can be used for $p>k.$ Your bound may be slightly better in localized cases where $p=k$.
> >
> > Zooming out, nearly optimal DP-PCA bounds in the spectral norm where already known. The main new contribution here is to get such bounds when the only thing we're allowed to do is add noise to the input matrix (i.e., input perturbation).  The present analysis has the advantage that it gives new perturbation bounds along the way that may be of independent interest.
> >
> > Todo items: Please verify the comparison with NPM, what you wrote about $\sigma_1$ looks wrong to me, but my quick check might also be wrong.
> >
> > To conclude, I think this is a good paper that should be accepted. I don't see a major breakthrough, though, since nearly optimal spectral norm bounds were already known for upwards of a decade.

---

> > > ### Author Response · Authors · 2025-08-06
> > >
> > > > The factor $\sigma_1$ that you put in the error bound for the noisy power method (NPM) doesn't seem quite right to me. [...] Specifically, Theorem 1.3 in NPM gives an error like (ignoring $\tilde{O}$, $\varepsilon$, $\delta$): $\frac{\sqrt{n}\sqrt{p}\sqrt{\sigma_k}}{\delta_k^{1.5}}, \dots$
> > >
> > > Thank you for your response and engagement during the rebuttal-discussion period. We appreciate the reviewer’s positive overall assessment of the paper and their recommendation for acceptance.
> > >
> > > We have carefully re-examined the work of Hardt and Price, and we appreciate the opportunity to clarify the comparison.
> > >
> > > There appears to be a misunderstanding. First, Theorem 1.3 of Hardt–Price (NeurIPS 2015) bounds a different quantity—**subspace error, not the spectral-norm error of a low-rank approximation**. Second, as explained in our earlier response, one can derive bounds for our setting from their work, but these are weaker than ours. We also want to highlight at the outset that our work resolves an open problem posed in Mangoubi–Vishnoi (J. ACM 2025) (link), giving the first high-probability spectral-norm bounds for $\\|\tilde{A}_p - A_p\\|$ in the input-perturbation model. For these reasons, we respectfully disagree with the statement that *“nearly optimal spectral-norm bounds were already known for upwards of a decade.”*
> > >
> > > We recall Theorem 1.3 from Hardt–Price for completeness ([arXiv link](https://arxiv.org/pdf/1311.2495)):
> > >
> > > ---
> > >
> > > **Theorem 1.3 (Hardt–Price, paraphrased)**
> > > Let $A \in \mathbb{R}^{n\times n}$ be symmetric with singular values $\sigma_1 \ge \dots \ge \sigma_n$, $U_p$ its top-$p$ singular vectors, and $X_L$ the output of NPM. For $k \ge p$, after
> > > $$
> > > L = O\\left( \frac{\sigma_p}{\sigma_p - \sigma_{p+1}} \log n \\right)
> > > $$
> > > iterations, with probability $9/10$,
> > > $$
> > > \\|(I - X_L X_L^\top) U_p\\| \le O \\left(  \frac{\sigma     \sqrt{n \log L}}{\sigma_p - \sigma_{p+1}} \cdot \frac{\sqrt{k}}{\sqrt{k} - \sqrt{p-1}}  \cdot \max_\ell \\|X_\ell \\| \tiny{\infty}  \\right),
> > > $$
> > > where $\sigma$ in this theorem (as defined in their Fig. 3) is the privacy-noise scale, roughly $\sqrt{pL}/\varepsilon$, not the top singular value $\sigma_1$ of $A$.
> > >
> > > This bound measures the **tangent of the principal angle** between $X_L$ and $U_p$; it is **not** a bound on $\\|\tilde{A}_p - A_p\\|$.
> > >
> > > ---
> > >
> > > **From subspace error to low-rank error**
> > > As noted in our earlier reply, one can project $A$ onto $X_L$ (Hardt–Price, Sec. 4.1) by setting $A' = X_L X_L^\top A$, yielding:
> > >
> > > $$
> > > \\|A' - A_p \\| \\leq \tilde{O} \\left( \frac{\sqrt{n \log(1/\delta)}}{\varepsilon} \cdot \frac{\sigma_1}{\delta_p} \cdot \max_{1 \leq i \leq n} \\|u_i\\| \tiny{\infty} \\right).
> > > $$
> > > This introduces $\sigma_1$ factor that is absent from the subspace-error bound above.
> > >
> > > ---
> > >
> > > **Our results (input-perturbation model)**
> > > We add symmetric Gaussian noise $E$ to $A$ and return $\tilde{A}_p$, the best rank-$p$ approximation to $\tilde{A} = A + E$. Under $\delta_p > 4\\|E\\| = C\sqrt{n}$, we prove:
> > >
> > > - **Theorem 2.1**:
> > >   $$
> > >   \\|\tilde{A}_p - A_p\\| = O\!\left( \frac{\sqrt{n \log(1/\delta)}}{\varepsilon} \cdot \frac{\sigma_p}{\delta_p} \right).
> > >   $$
> > >
> > > - **Theorem 2.2**:
> > >   $$
> > >   \\|\tilde{A}_p - A_p\\| = \tilde{O}\!\left( \frac{\sqrt{n \log(1/\delta)}}{\varepsilon} + \frac{r^2 \sigma_p}{\delta_p} \right),
> > >   $$
> > >   where $r \ge p$ is the smallest index such that $\sigma_r \le \sigma_p / 2$.
> > >
> > > ---
> > >
> > > **Improvements over projected-NPM**
> > > Our bounds can improve over the projected-NPM bound by up to $\sqrt{n}$ in common structured regimes:
> > >
> > > - **Localized eigenvectors**: Theorem 2.1 gains a factor $\tilde{O}(\sigma_1/\sigma_p)$, up to $\sqrt{n}$ when $\sigma_1 = \Theta(n)$ and $\sigma_p = \Theta(\sqrt{n})$.
> > > - **Low stable rank**: Theorem 2.2 gains a factor $\min\{\frac{\sqrt{n}}{r^2}, \frac{\sigma_1}{\delta_p}\}$, up to $\sqrt{n}$ when $r = \tilde{O}(1)$ and $\delta_p = \Theta(\sqrt{n})$.
> > >
> > > When **all** eigenvectors are fully delocalized, Theorem 2.1 matches projected-NPM in the typical $\sigma_1/\sigma_p = \Theta(\sqrt{n})$ regime, and Theorem 2.2 still improves if $r = \tilde{O}(1)$.
> > >
> > >
> > > ---
> > >
> > > **Conclusion**
> > > Theorem 1.3 in Hardt–Price and our Theorems 2.1–2.2 bound fundamentally different quantities in different models. To the best of our knowledge—our work provides the **first high-probability spectral-norm bounds** for $\\|\tilde{A}_p - A_p\\|$ in the input-perturbation model.

---

> > > > ### Comment · Reviewer_aeeH · 2025-08-06
> > > >
> > > > You're right. The factor $\sigma_1$ has to come in when converting the subspace guarantee to the low-rank approximation.
> > > >
> > > > Thanks for the thorough discussion. This helped me appreciate the significance of the result better. I'll raise my score accordingly.

---

### Official Review · Reviewer_DQsb · 2025-07-04

**Clarity:** 3
**Significance:** 3
**Originality:** 3
**Rating:** 5
**Confidence:** 4

**Summary:**

This paper develops general perturbation bounds for spectral functions of a matrix. Specifically, for a matrix $A$, with eigenvalues and eigenvectors $\{(\lambda_i, u_i): i = 1,2, \dots N\}$, let $f_p(A)$ denote the matrix $\sum_{i=1}^p f(\lambda_i) u_i u_i^\top$. The main result of the paper provides a general purpose bound on $|| f(A+E) - f(A)||$ in terms $||E||$, where $E$ denotes a perturbation in the matrix.

As a primary application of their result, the authors analyze the utility (as measured by the operator norm error between the privatized best rank-$p$ approximation and the actual best rank-$p$ approximation)  for the Gaussian mechanism in differentially private PCA, improving existing results.

**Questions:**

Questions and Comments:
— I think the authors should discuss prior work on deriving perturbation bounds using the contour integral representation more prominently. Currently, those citations are easy to miss (in step 1 of the proof outline) and this sends the misleading impression that this technique is developed in this paper. It would be ideal to discuss these prior works as early as possible and in some detail.

— Is it possible to state the bound in Theorem 2.3 with explicit constants? This would make the result user-friendly. Currently, it might not be very obvious to a reader what the constants hidden in the big-O notation depend on (without carefully reading the proof).

— In the “Empirical Results” section the authors claim the bound $7 ||E|| \cdot \tfrac{\lambda_p}{\delta_p}$.  Where does the constant $7$ come from? (The results in the main paper are stated in big-O notation).

— Does the bound apply to any submultiplicative norm, or is there something special about the operator norm? If the bound is general, it would be good to state it generally.

**Ethical Concerns:**

["NO or VERY MINOR ethics concerns only"]

**Final Justification:**

This paper develops perturbation bounds for spectral functions of a matrix using a contour integral representation of spectral functions. The bounds are very general and can have many potential applications. Because of this reason, I think the paper definitely meets the bar of acceptance.

During the discussion period, the authors addressed all of my suggestions, including stating the bounds with explicit constants so that it is easy to apply.

**Limitations:**

Yes

**Quality:**

3

**Strengths And Weaknesses:**

Strengths:
— The paper is well-written and the authors have made a good effort to present their proofs in an accessible and systematic fashion.
— The bound appears to be quite general, and could have many potential applications.
—The authors use a contour integral representation of spectral functions in terms of the resolvent to obtain their perturbation bound. This technique is well-known and classical in matrix perturbation theory. The primary contribution of this paper is to exploit this technique to develop a general-purpose bound which can be readily applied without a case-by-case analysis of the contour integral representation. To the best of my knowledge, I am not aware of a similar general purpose bound, so this contribution appears novel, although the contour integral technique is quite standard.

Weaknesses:
— The authors consider a single application of their bound, and moreover, the improvement in the application is not very significant (an operator norm bound rather than the Frobenius norm bound). Some additional applications could have improved the paper.

---

> ### Author Rebuttal · Authors · 2025-07-30
>
> Thank you for your valuable comments and suggestions. We are glad that you appreciate our new bounds, the novelty of the contour bootstrapping argument, and the potential applications of our work. We address your specific questions and concerns below.
> ### Questions:
> >Is it possible to state the bound in Theorem 2.3 with explicit constants?
>
> Thank you for the question. The constant in Theorem 2.3 is 4. This value can be further optimized by examining the detailed computation in Appendix E. We will update Theorem 2.3 accordingly in the final version.
>
> >In the “Empirical Results” section, the authors claim the bound $7\\|E\\| \lambda_p/\delta_p$. Where does the constant 7 come from?
>
> We appreciate the opportunity to clarify.
> The constant $7$ comes from the exact r.h.s of Theorem 2.1, which is $6 \\|E\\| ( \log (6\frac{\lambda_p}{\delta_p}) + \frac{\lambda_p}{\delta_p})$. Since the logarithmic term is negligible compared to $\frac{\lambda_p}{\delta_p}$ in our real-world datasets (as considered in this section), we opted to simplify the bound by replacing the constant 6 with the slightly larger integer 7.
>
> We will add this clarification in Section 4 of the final version.
>
> > Does the bound apply to any submultiplicative norm, or is there something special about the operator norm?
>
> Thank you for the question. Our bound is specifically tailored to the operator norm, and does not directly extend to general submultiplicative norms.
>
> The key step in estimating the term $F_1$ (see lines 247–265) crucially relies on properties that are specific to the operator norm. In particular, we use the identity:
> $$
> \left\\| (zI - A)^{-1} \right\\| = \left\\| \sum_{i=1}^n \frac{u_i u_i^\top}{z - \lambda_i} \right\\| \leq \max_{1 \leq i \leq n} \frac{1}{|z - \lambda_i|},
> $$
> and
> $$
> \left\\| \sum_{i > r} \frac{u_i u_i^\top}{z - \lambda_i} \right\\| \leq \max_{i > r} \frac{1}{|z - \lambda_i|},
> $$
> which are sharp in the operator norm but do not hold analogously for other submultiplicative norms such as the Frobenius norm or Schatten-$p$ norms.
>
> That said, the contour bootstrapping framework itself can, in principle, be adapted to other norms. However, obtaining sharp analogues of our bounds in those settings would require new norm-specific estimates of resolvent decay and perturbation alignment. We view this as a promising direction for future work and will include a brief remark in the final version to clarify this scope.
>
> ### Other comments:
>
> >The authors consider a single application of their bound, and moreover, the improvement in the application is not very significant (an operator norm bound rather than the Frobenius norm bound). Some additional applications could have improved the paper.
>
> Thank you for the comment. While our method has broader applicability, we chose to focus on a single motivating application—differentially private PCA—to present the technique and its implications with greater clarity. We will consider highlighting potential extensions more explicitly in the revision.
>
> >I think the authors should discuss prior work on deriving perturbation bounds using the contour integral representation more prominently.  ... It would be ideal to discuss these prior works as early as possible and in some detail.
>
> Thank you for these valuable comments. We fully acknowledge that the contour integral representation is a classical tool, widely used in prior work on functional perturbation analysis.
>
> Our contribution does not lie in introducing this technique, but in using it as the foundation for a novel and general-purpose bootstrapping strategy that applies broadly to spectral-norm perturbation bounds.
>
> We will revise the introduction and related work sections to include a more prominent and timely discussion of classical uses of the contour method. In particular, we will clarify this distinction in Steps 1 and 2 of the “Outline of Proof.” To our knowledge, most prior applications of contour methods focus on eigenvalue or eigenspace perturbations (e.g., Davis–Kahan-type results), which differ from our setting of low-rank approximation under spectral norm.

---

> > ### Comment · Reviewer_DQsb · 2025-08-06
> >
> > Thank you to the authors for their detailed response to my questions. I continue to have a very positive view of this paper.

---

### Decision · Program_Chairs · 2025-09-17

**Decision:**

Accept (oral)

**Comment:**

The paper introduce a new perturbation bounds for low rank approximations in the spectral norm, improving the Eckart-Young-Mirsky theorem. Then it presents a practical application to differentially private PCA.

From a technical perspective the paper is novel and interesting. More specifically, the authors have a new technique that avoids some of the classical tools like Davis-Kahan expansions in the proof of Eckart-Young-Mirsky theorem and do they are able to obtain tighter extensions to classical results from numerical analysis and matrix theory.

From a practical perspective, the authors shows how these new results can be applied to very practical problems as differentially private PCA.

From exposition perspective, the paper is very accessible and easy to read.

Overall, the reviewers appreciate the novelty and the strength of the new result and its applicability in differential privacy, overall the paper is strong and should be accepted.